# Risk-Bounded Distribution Reconstruction: Stable Statistic Calibration for Long-Tailed Recognition

Guanliang Liu [1]    Wenchao Chen [1]    Long Tian [2]    Xuefei Cao [3]    Hongwei Liu [1]    Bo Chen [1]

## Abstract

Long-tailed recognition suffers from extreme class imbalance, where scarce tail data leads to biased and fragile feature distributions that exacerbate confusion with semantically or visually similar classes. Prior feature-space reconstruction methods transfer head-class structure or train conditional generators to synthesize tail features, yet the resulting *statistical* updates are often heuristic and can degrade multi-class separability when tail estimates are unreliable. Given this issue, we propose *Risk-Bounded Distribution Reconstruction* (RBDR), an offline statistic calibration framework for the two-stage long-tailed pipeline, grounded in an analysis of rival-induced discriminative directions. RBDR performs (i) risk-aware mean calibration by softly projecting any candidate update onto a supportive set such that a surrogate discriminative-risk upper bound does not increase, and (ii) covariance control by shrinking dispersion in a rival subspace while preserving diversity in orthogonal components. In the frozen-feature reconstruction setting, these calibration components convert heuristic reconstruction cues into controllable distributional updates, improving both performance and stability across multiple long-tailed benchmarks.

## 1. Introduction

Long-tailed recognition suffers from extreme class imbalance: head classes are data-rich while tail classes are severely under-sampled (Yang et al., 2022; Zhang et al.,

[1]National Key Laboratory of Radar Signal Processing, Xidian University, Xi'an, Shaanxi, China [2]School of Computer Science and Technology, Xidian University, Xi'an, Shaanxi, China [3]School of Cyber Engineering, Xidian University, Xi'an, Shaanxi, China. Correspondence to: Bo Chen <bchen@mail.xidian.edu.cn>, Wenchao Chen <wechen_xidian@163.com>.

*Proceedings of the $43^{rd}$ International Conference on Machine Learning*, Seoul, South Korea. PMLR 306, 2026. Copyright 2026 by the author(s).

2023). With limited tail observations, the learned representations capture an incomplete and biased view of tail feature distributions (Yi et al., 2025). This incomplete supervision leads to fragile tail geometry: class prototypes drift, intra-class structure is distorted, and tail samples are more likely to be misclassified into visually or semantically similar classes (Yang et al., 2021; Guo et al., 2022b; Chen & Su, 2023).

A promising direction is feature-space tail distribution reconstruction, which compensates for missing tail information by transferring reliable structures from head classes or auxiliary datasets, and then synthesizing additional tail features for subsequent classifier fine-tuning. Existing methods realize this idea in several ways: transferring head-class geometric cues (e.g., principal directions or eigenspectra) (Liu et al., 2021; Ma et al., 2024; 2025); calibrating tail moments using head statistics (e.g., means and covariances) (Wang et al., 2022; Ma et al., 2023); and augmenting tail data via class-conditional generation (e.g., diffusion models) (Han et al., 2024; Shao et al., 2024; Liang et al., 2025) or head-to-tail feature blending (Li et al., 2024; 2025a). Despite strong empirical progress, most reconstruction rules remain largely heuristic: they prescribe how to update tail statistics but provide limited theoretical justification on whether such updates are discriminatively safe under unreliable tail estimates.

A central challenge in long-tailed recognition is preserving multi-class separability (Menon et al., 2020). Errors are not evenly distributed: tail classes are often confused with a limited number of highly similar rivals (Wang et al., 2022; Ma et al., 2023). Under scarce data, estimating the separation against these rivals is unreliable, so unconstrained calibration can inadvertently shift or inflate the tail distribution along rival-aligned discriminative directions—potentially matching tail moments better yet worsening confusion in the overall label space (Zhao et al., 2026). This motivates our perspective: we analyze reconstruction through *rival directions*, induced by each tail class and its most confusable opponents, and ask whether tail statistics can be calibrated while controlling the increase of rival confusion.

To this end, we propose *Risk-Bounded Distribution Reconstruction* (RBDR), a framework for safe and controllable

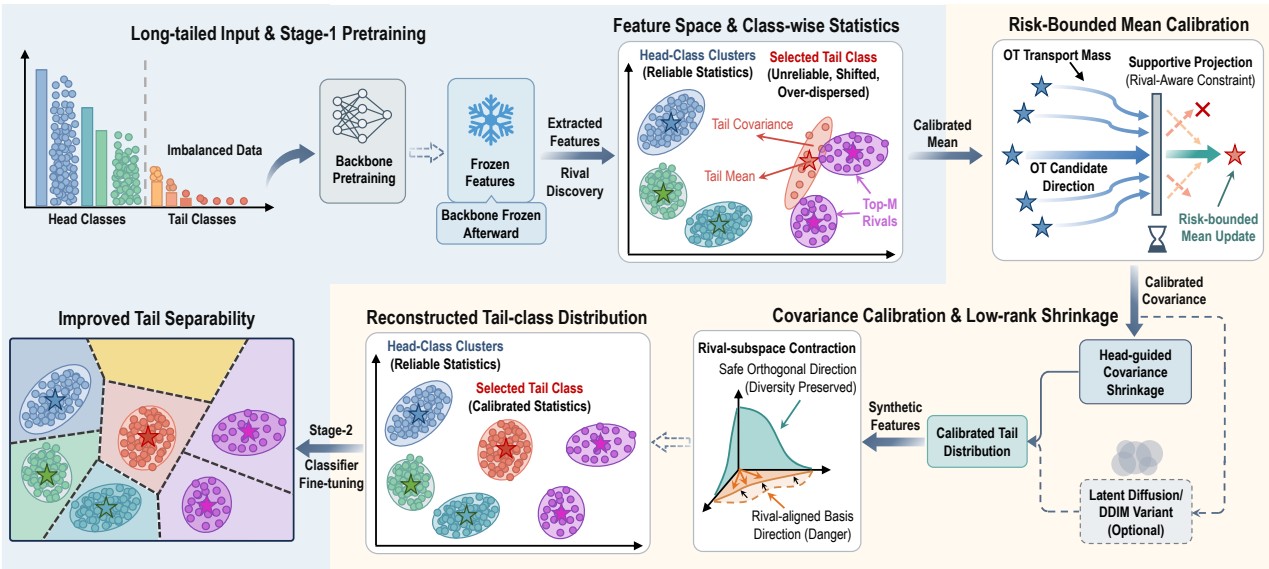

*Figure 1.* Overview of RBDR. After Stage-1 training, the backbone is frozen and class-wise statistics are estimated in the feature space. For each tail class, RBDR identifies top-$M$ rivals, obtains an OT-based head-to-tail candidate direction, performs rival-aware supportive projection for risk-bounded mean calibration, calibrates covariance via head-guided shrinkage, applies low-rank rival-subspace contraction, and finally uses the reconstructed tail distributions for Stage-2 classifier fine-tuning.

offline statistic calibration in long-tailed recognition. We consider the common two-stage pipeline: a backbone is trained first, class-wise statistics are estimated on frozen features, and tail distributions are reconstructed offline to synthesize tail features for classifier fine-tuning. While offline calibration is explicit and inspectable, inaccurate updates can be systematically amplified during fine-tuning. RBDR addresses this by imposing risk-aware constraints derived from a rival-direction discriminative analysis. For each tail class, we (i) characterize a supportive set of mean-update directions under which a surrogate discriminative risk bound is non-increasing, and obtain a practical update by softly projecting any candidate direction (e.g., a direction suggested by similarity-based head-to-tail transfer) onto this set; and (ii) identify a dangerous subspace spanned by rival directions and perform low-rank covariance shrinkage to prevent harmful dispersion along these directions while preserving diversity in orthogonal components. Together, these steps convert heuristic reconstruction signals into principled modular calibration operators, providing a controllable offline refinement for standard two-stage, frozen-feature reconstruction pipelines.

Our contributions are summarized as follows:

- For long-tailed recognition, we establish a rival-direction-based analysis for offline feature-space tail reconstruction and derive a risk-bounded theoretical framework via a threshold-optimized Cantelli surrogate, which links statistic updates to changes in multi-class confusion.

- We propose RBDR, which instantiates head-to-tail reconstruction cues with an Optimal Transport (OT) initialization (Cuturi, 2013) and turns them into bounded calibration via (i) supportive projection for mean updates and (ii) low-rank shrinkage in rival-induced subspaces to suppress harmful dispersion.

- Experiments on multiple long-tailed benchmarks show consistent gains. Ablations on module design, initialization choices, hyperparameters, and computational cost validate the contribution of each component and the practicality of the overall framework.

## 2. Risk-Bounded Analysis for Offline Tail Reconstruction

### 2.1. Preliminaries

**Long-tailed classification.** Let $\mathcal{D} = \{(I_i, y_i)\}_{i=1}^{N}$ be a long-tailed training set with $K$ classes, where $y_i \in [K] := \{1, \ldots, K\}$ and $N = \sum_{k=1}^{K} n_k$ with $n_k := |\{i : y_i = k\}|$. Without loss of generality, assume $n_1 \geq n_2 \geq \cdots \geq n_K$. Let $\mathcal{C}_{\text{tail}} \subseteq [K]$ denote the set of tail classes (e.g., those with small $n_k$).

**Pretraining and frozen-feature setting.** We first train a feature extractor $\phi(\cdot; \theta)$ and a linear classifier $W = [w_1, \ldots, w_K] \in \mathbb{R}^{d \times K}$ by minimizing the cross-entropy loss:

$$\min_{\theta, W} \mathbb{E}_{(I,y) \sim \mathcal{D}} \left[ \mathcal{L}_{\text{CE}}\left(W^{\top} \phi(I; \theta), y\right) \right], \quad (1)$$

where $\phi(I; \theta) \in \mathbb{R}^d$ is the feature vector and $W^{\top} \phi(I; \theta) \in$

$\mathbb{R}^K$ are the logits (bias omitted for simplicity). After training, we freeze $\phi$ and denote the extracted feature of input $I$ by $x = \phi(I) \in \mathbb{R}^d$.

**Class-wise statistics.** Given frozen features $\{x_i\}_{i:y_i=c}$ with $n_c$ samples in class $c$, we compute

$$\hat{\mu}_c := \frac{1}{n_c} \sum_{i:y_i=c} x_i, \quad \hat{\Sigma}_c := \frac{1}{n_c-1} \sum_{i:y_i=c} (x_i - \hat{\mu}_c)(x_i - \hat{\mu}_c)^\top. \tag{2}$$

In long-tailed settings, $\hat{\mu}_c$ and $\hat{\Sigma}_c$ can be unreliable for tail classes due to small $n_c$.

**Rival classes.** Tail errors typically concentrate on a small set of confusing opponents. For each tail class $c \in \mathcal{C}_{\text{tail}}$, we define a rival set $\mathcal{K}_c \subseteq [K] \setminus \{c\}$ (e.g., the top-$M$ most confusable classes under the frozen classifier), with weights $\{\omega_{c,k}\}_{k\in\mathcal{K}_c}$ satisfying $\omega_{c,k} \geq 0$ and $\sum_{k\in\mathcal{K}_c} \omega_{c,k} = 1$.

## 2.2. A Distribution-Free Pairwise Confusion Upper Bound

In long-tailed recognition, improving a tail class often amounts to reducing its confusion with a small set of rivals. However, an overly aggressive correction may also increase the reverse errors (i.e., rival-to-tail confusion), which can harm overall performance. Motivated by this trade-off, we adopt a pairwise view: fix a tail class $c$ and one rival $k \in \mathcal{K}_c$. Consider a one-vs-one linear rule parameterized by a projection direction $w \in \mathbb{R}^d$ and threshold $t \in \mathbb{R}$: predict $c$ if $w^\top x \geq t$, and predict $k$ otherwise. Let $Z := w^\top x$. Let $\pi_c, \pi_k$ denote class priors (or re-weighted priors), and let $C_{c,k}, C_{k,c} > 0$ denote misclassification costs.

**Definition 2.1.** (Cost-sensitive pairwise risk). The weighted pairwise confusion risk is defined as:

$$\begin{aligned} R_{c,k}(t,w) := \; &\pi_c C_{c,k} \mathbb{P}(Z < t \mid y = c) \\ &+ \pi_k C_{k,c} \mathbb{P}(Z \geq t \mid y = k). \end{aligned} \tag{3}$$

To obtain a closed-form bound for (3) without modeling $x \mid y$, we exploit that classification depends on the scalar projection $Z = w^\top x$. The two error terms in (3) thus become one-sided tail probabilities of $Z \mid y$. Although the full conditional distributions are unknown, their first and second moments are estimable, which motivates the following moment-only assumption and a distribution-free surrogate bound.

**Assumption 2.2.** For each $y \in \{c, k\}$, the conditional distribution $x \mid y$ has finite mean and covariance.

**Definition 2.3.** (Projected moments). Let $\mu_y := \mathbb{E}[x \mid y]$ and $\Sigma_y := \text{Cov}(x \mid y)$. For any $w \in \mathbb{R}^d$, define: $m_y(w) := \mathbb{E}[Z \mid y] = w^\top \mu_y, s_y^2(w) := \text{Var}(Z \mid y) = w^\top \Sigma_y w$. For a threshold $t \in \mathbb{R}$, define the signed distances: $d_c := m_c(w) - t, d_k := t - m_k(w)$. In practice, we use the empirical estimates $(\hat{\mu}_y, \hat{\Sigma}_y)$ as plug-in approximations.

**Cantelli inequality.** We use the standard Cantelli inequality (Ghosh, 2002): for any real-valued $X$ with finite mean and variance and any $a > 0$:

$$\mathbb{P}(X - \mathbb{E}[X] \geq a) \leq \frac{\text{Var}(X)}{\text{Var}(X) + a^2}. \tag{4}$$

Applying (4) to the two one-sided tail events in Definition 2.1, we obtain the following distribution-free surrogate.

**Lemma 2.4.** *(Cantelli surrogate for pairwise risk). Under Assumption 2.2, the risk in* (3) *satisfies:*

$$\begin{aligned} R_{c,k}(t,w) \leq \overline{R}_{c,k}(t,w) := \; &\pi_c C_{c,k} \frac{s_c^2(w)}{s_c^2(w) + d_c^2} \\ &+ \pi_k C_{k,c} \frac{s_k^2(w)}{s_k^2(w) + d_k^2}, \end{aligned} \tag{5}$$

*Proof in Appendix B.1.*

To make the bound interpretable in terms of inter-class separation and to simplify threshold selection, we center $t$ at the midpoint of the two projected means and use a scalar offset $u$ for the remaining freedom. This symmetric parameterization isolates the effect of the projected gap and enables us to eliminate the threshold by defining a threshold-optimized surrogate via $\min_u$, which we use throughout the subsequent analysis.

**Definition 2.5.** (Centered threshold parameterization and threshold-optimized surrogate). Define the projected mean gap:$\Delta(w) := m_c(w) - m_k(w) = w^\top(\mu_c - \mu_k)$, and parameterize the threshold by: $t = \frac{m_c(w)+m_k(w)}{2} + u, u \in \mathbb{R}$. Then $d_c = \frac{\Delta(w)}{2} - u$ and $d_k = \frac{\Delta(w)}{2} + u$. Define:

$$\begin{aligned} F\big(\Delta(w), s_c^2(w), s_k^2(w), u\big) := \; &\pi_c C_{c,k} \frac{s_c^2(w)}{s_c^2(w) + \left(\frac{\Delta(w)}{2} - u\right)^2} \\ &+ \pi_k C_{k,c} \frac{s_k^2(w)}{s_k^2(w) + \left(\frac{\Delta(w)}{2} + u\right)^2}, \end{aligned} \tag{6}$$

and the *threshold-optimized surrogate*:

$$G\big(\Delta(w), s_c^2(w), s_k^2(w)\big) := \min_{u \in \mathbb{R}} F\big(\Delta(w), s_c^2(w), s_k^2(w), u\big). \tag{7}$$

Definition 2.5 cleanly separates decision-side parameters from distribution-side quantities. For fixed class statistics $(\mu_c, \Sigma_c)$ and $(\mu_k, \Sigma_k)$, the decision rule is parameterized by $(w, u)$, whereas the statistics enter the surrogate only through $\Delta(w)$ and $s_c^2(w), s_k^2(w)$. Although $w$ and $u$ are typically learned jointly and thus depend implicitly on the statistics, it is convenient to eliminate the threshold by defining the optimized surrogate $G(\cdot) = \min_u F(\cdot)$. This removes the remaining degree of freedom in $t$ and allows us to analyze the bound directly in terms of $\Delta(w)$ and $s_c^2(w), s_k^2(w)$.

**Definition 2.6.** (Pairwise Fisher ratio). Let $S_w := \Sigma_c + \Sigma_k \succeq 0$. Define:

$$J(w) := \frac{\Delta(w)^2}{w^\top S_w w} = \frac{\left(w^\top (\mu_c - \mu_k)\right)^2}{w^\top (\Sigma_c + \Sigma_k) w}. \quad (8)$$

We further define the optimal Fisher separation by $J^\star := \max_w J(w)$. When $S_w \succ 0$, $J^\star = (\mu_c - \mu_k)^\top S_w^{-1}(\mu_c - \mu_k)$.

**Theorem 2.7.** *(Surrogate–Fisher consistency). Consider the threshold-optimized Cantelli surrogate $G(\Delta(w), s_c^2(w), s_k^2(w))$ and the Fisher ratio $J(w)$.*

*(i) For fixed projected variances $(s_c^2(w), s_k^2(w))$, the optimized surrogate $G(\Delta(w), s_c^2(w), s_k^2(w))$ is non-increasing in $|\Delta(w)|$.*

*(ii) Maximizing $J(w)$ is aligned with maximizing $|\Delta(w)|$, and therefore favors smaller values of $G$.*

*Proof in Appendix B.2.*

**Role of the surrogate.** The Cantelli surrogate is generally conservative and not intended to be a tight estimate of the true confusion risk. We do not optimize this bound; instead, we use its monotone relation to Fisher separation to derive *directional* and *safe* statistic-calibration rules under unreliable tail estimates.

Theorem 2.7 establishes an envelope-style link between the optimized Cantelli surrogate and Fisher separation: after eliminating the threshold via $\min_u$, the surrogate decreases as the projected gap $|\Delta(w)|$ grows (i), and optimizing the Fisher ratio promotes larger gaps (ii). This motivates expressing the best achievable separation through $J^\star$ and treating the decision parameters as implicitly optimized, so that we can focus on how statistic calibration changes $(\mu, \Sigma)$ and hence $J^\star$ (and $G$) without differentiating through the implicit optimizers $w^\star(\mu, \Sigma)$ and $u^\star(\mu, \Sigma)$. This viewpoint aligns with our calibrate-then-finetune workflow: the offline calibration stage updates $(\mu, \Sigma)$ while keeping the current $(w, u)$ fixed, and the subsequent fine-tuning stage updates $(w, u)$ with $(\mu, \Sigma)$ held fixed.

### 2.3. Monotone Surrogate Improvement via Statistic Calibration

Building on the surrogate–Fisher link in Section 2.2, we study how calibrating class statistics changes the optimal Fisher separation $J^\star$ and thereby affects the optimized surrogate $G$.

**Lemma 2.8.** *(Mean calibration increases $J^\star$ under a sufficient condition). Consider a mean shift $\mu_c(\alpha) = \mu_c + \alpha\delta$ for $\alpha \geq 0$, keeping $(\Sigma_c, \Sigma_k, \mu_k)$ fixed. Let $d := \mu_c - \mu_k$ and $S_w \succ 0$. Then:*

$$J^\star(\alpha) = (d + \alpha\delta)^\top S_w^{-1}(d + \alpha\delta). \quad (9)$$

*If $\delta^\top S_w^{-1} d > 0$, then $J^\star(\alpha)$ is strictly increasing in $\alpha$ for all $\alpha \geq 0$, and consequently $G$ is non-increasing by Theorem 2.7. Proof in Appendix B.3.*

**Lemma 2.9.** *(Covariance calibration increases $J^\star$ on the admissible range). Consider $\Sigma_c(\beta) = \Sigma_c - \beta Q$ with $\beta \geq 0$ and $Q \succeq 0$, keeping $(\mu_c, \mu_k, \Sigma_k)$ fixed. Let $d := \mu_c - \mu_k$ and $S_w(\beta) := \Sigma_c(\beta) + \Sigma_k = S_w - \beta Q$, where $S_w := \Sigma_c + \Sigma_k \succ 0$. Define:*

$$\beta_{\max} := \frac{1}{\lambda_{\max}\left(S_w^{-1/2} Q S_w^{-1/2}\right)} \in (0, \infty], \quad (10)$$

*Then for any $\beta \in [0, \beta_{\max})$, $S_w(\beta) \succ 0$ and $J^\star(\beta) = d^\top S_w(\beta)^{-1} d$ is non-decreasing in $\beta$, hence $G$ is non-increasing by Theorem 2.7. Proof in Appendix B.4.*

Intuitively, the condition $\delta^\top S_w^{-1} d > 0$ indicates that the mean update moves class $c$ away from its rival $k$ in a Fisher-preferred direction, thereby increasing the optimal separation $J^\star$. With scarce tail samples, covariance estimates can be highly uncertain and may overestimate within-class spread, which increases overlap with rival classes. Consequently, any admissible shrinkage that reduces within-class scatter yields a non-decreasing $J^\star(\beta) = d^\top S_w(\beta)^{-1} d$, and thus a non-increasing threshold-optimized surrogate $G$.

### 2.4. Multi-Rival Aggregation

So far we have focused on a fixed tail–rival pair $(c, k)$. In multi-class long-tailed recognition, however, tail errors typically concentrate on a *set* of rivals. We therefore aggregate pairwise quantities over $\mathcal{K}_c$ using the nonnegative weights $\{\omega_{c,k}\}_{k \in \mathcal{K}_c}$ with $\sum_k \omega_{c,k} = 1$ (see Section 2.1).

**Definition 2.10.** *(Aggregated objectives over rivals). For a tail class $c$ with rival set $\mathcal{K}_c$ and weights $\{\omega_{c,k}\}_{k \in \mathcal{K}_c}$, define the aggregated threshold-optimized surrogate and the aggregated optimal Fisher separation as:*

$$\bar{G}_c := \sum_{k \in \mathcal{K}_c} \omega_{c,k} G_{c,k}, \quad \bar{J}_c^\star := \sum_{k \in \mathcal{K}_c} \omega_{c,k} J_{c,k}^\star, \quad (11)$$

*where $G_{c,k}$ denotes the pairwise surrogate for the pair $(c, k)$, and $J_{c,k}^\star := \max_w J_{c,k}(w)$ is the corresponding optimal Fisher value.*

The following observation lifts the pairwise monotonicity results to the aggregated objectives.

**Proposition 2.11.** *(Aggregated monotonicity). If, along a calibration path, $J_{c,k}^\star$ is non-decreasing for every $k \in \mathcal{K}_c$, then $\bar{J}_c^\star$ is non-decreasing and $\bar{G}_c$ is non-increasing. A sufficient condition for this is: (i) mean calibration $\mu_c(\alpha) = \mu_c + \alpha\delta \ (\alpha \geq 0)$ whenever $\delta^\top (\Sigma_c + \Sigma_k)^{-1}(\mu_c - \mu_k) > 0$ for all $k \in \mathcal{K}_c$ (Lemma 2.8), and for (ii) covariance calibration $\Sigma_c(\beta) = \Sigma_c - \beta Q \ (Q \succeq 0)$ whenever $\beta \in [0, \bar{\beta}_{\max})$ with $\bar{\beta}_{\max} := \min_{k \in \mathcal{K}_c} \beta_{\max}^{(k)}$ so that $(\Sigma_c - \beta Q) + \Sigma_k \succ 0$ for all $k$ (Lemma 2.9). Proof in Appendix B.5.*

Proposition 2.11 provides a simple bridge from pairwise analysis to multi-rival behavior: it is sufficient to ensure non-decreasing separation against each rival to improve the aggregated objectives. In the multi-rival setting, mean calibration is typically more restrictive than covariance calibration under our conservative sufficient condition: a single direction $\delta$ must satisfy all pairwise constraints for every $k \in \mathcal{K}_c$, as different rivals generally favor different optimal directions, making cross-rival trade-offs difficult to characterize. By contrast, covariance shrinkage only requires selecting a step size $\beta$ in the intersection of per-rival feasible ranges (e.g., $\beta < \min_{k \in \mathcal{K}_c} \beta_{\max}^{(k)}$).

The analysis above finally provides a diagnostic view of existing distribution reconstruction methods and reveals the direction of method design: when they help and where they can break.

(i) *Covariance-only* recalibration (Ma et al., 2023; 2024; 2025) borrows more reliable head-class geometric structure to regularize tail dispersion while keeping the tail mean fixed. Operationally, this corresponds to shrinking the tail covariance as $\Sigma_c(\beta) = \Sigma_c - \beta Q$, which makes tail features more compact and reduces confusion with rivals, thereby decreasing the surrogate bound (Proposition 2.11). Its limitation is that the tail mean remains unchanged, so substantial overlap may persist when the tail class is already close to its rivals.

(ii) Methods that recalibrate *both* mean and covariance (Wang et al., 2022; Chen & Su, 2023) often use a head-aggregated mean shift over the top-$k$ heads, e.g., $\delta = \sum_{i=1}^{k} \pi_{c,h_{(i)}} \left( \mu_{h_{(i)}} - \mu_c \right)$. Such updates help only when the induced direction is supportive with respect to the dominant rivals; otherwise, if the aggregate aligns with rival-induced directions (e.g., because some high-weight heads may also be confusable rivals), the mean may be pulled toward rivals and separability can deteriorate despite improved within-class fit.

# 3. Risk-Bounded Distribution Reconstruction

Section 2 establishes a monotone link from statistic calibration to a decrease of the aggregated surrogate $\bar{G}_c$. We now instantiate this principle as *Risk-Bounded Distribution Reconstruction* (RBDR): a safe and controllable offline calibration of tail-class means and covariances under rival-aware constraints.

## 3.1. Risk-Bounded Mean Calibration

**Head-to-tail candidate direction via optimal transport.** For each tail class $c \in \mathcal{C}_{\text{tail}}$, we construct an initial mean-update direction by transferring reliable head-class means. Rather than selecting a fixed neighbor set or using hand-crafted weights (e.g., top-$k$ nearest heads), we compute head-to-tail transfer weights via *optimal transport* (OT). OT provides a distribution-level mechanism to adaptively allocate transfer mass across all head classes through a globally coupled matching, avoiding commitment to a preset subset.

Concretely, we form discrete distributions over head and tail class means: $P = \sum_{h \in \mathcal{C}_{\text{head}}} p_h \, Dirac(\mu_h)$, $Q = \sum_{c \in \mathcal{C}_{\text{tail}}} q_c \, Dirac(\mu_c)$, where $Dirac(\cdot)$ denotes the Dirac measure and $(p_h, q_c)$ are nonnegative weights. We then solve the OT problem $T^\star \in \arg\min_{T \in \Pi(P,Q)} \langle T, C \rangle_F$, using Sinkhorn (Cuturi, 2013), where $\Pi(P,Q) := \Big\{ T \geq 0 \, \Big| \, \sum_{c \in \mathcal{C}_{\text{tail}}} T_{h,c} = p_h, \sum_{h \in \mathcal{C}_{\text{head}}} T_{h,c} = q_c \Big\}$ and $C$ is the cost matrix defined by distances between means. The resulting OT-based candidate direction for class $c$ is a convex combination of head-to-tail displacements:

$$\delta^{OT} := \sum_{h \in \mathcal{C}_{\text{head}}} \pi_{c,h} \left( \mu_h - \mu_c \right), \ \ \pi_{c,h} := \frac{T^\star_{h,c}}{\sum_{h'} T^\star_{h',c}}. \ (12)$$

**Supportive projection for multi-rival safety.** The OT direction $\delta^{OT}$ aggregates displacements from *all* head classes, and may therefore include components aligned with confusable rivals. We filter out such potentially harmful components while retaining the head-guided signal by projecting $\delta^{OT}$ onto the multi-rival supportive set:

$$\begin{aligned} \delta^\star := \arg\min_\delta \ &\| \delta - \delta^{OT} \|_2^2 \\ \text{s.t.} \quad &\delta^\top S_{c,k}^{-1} (\mu_c - \mu_k) \geq 0, \ \ \forall k \in \mathcal{K}_c. \end{aligned} \ (13)$$

We call the solution $\delta^\star$ the *supportive direction*. Intuitively, (13) makes the smallest adjustment to the OT direction, thereby retaining as much head-class information as possible, while ensuring that the update does not move the tail mean toward its confusable rivals. In practice, multi-rival constraints can be conservative under noisy tail statistics or disagreeing rivals; we therefore use a soft-penalty solver for (13) by default. Details of the hard and soft variants are provided in Appendices C.1 and C.2. When the constraints are highly restrictive (e.g., $\delta^{OT}$ conflicts with multiple rivals), the resulting $\delta^\star$ can be very small (possibly near-zero), yielding a conservative mean update that serves as a safeguard. We then update the tail mean as:

$$\mu_c \leftarrow \mu_c + \alpha \, \delta^\star. \ (14)$$

## 3.2. Covariance Calibration and Rival-Direction Control

While the analysis suggests that shrinking covariance can be beneficial, in practice the optimal shrinkage direction $Q$ is unknown. We therefore instantiate covariance calibration with an efficient closed-form head-guided shrinkage, while

also studying a class-conditional latent diffusion backend to explore more flexible feature synthesis.

**(i) Head-guided covariance shrinkage.** Tail covariances are often over-dispersed under scarce samples, while head covariances are typically more stable and tighter due to abundant data (see Figure 3). Let $\{\pi_{c,h}\}_{h \in \mathcal{C}_{\text{head}}}$ be the head-to-tail transport weights obtained in (12). We form a head-aggregated covariance proxy $\Sigma_c^{\text{H}} := \sum_{h \in \mathcal{C}_{\text{head}}} \pi_{c,h} \Sigma_h$, and calibrate the tail covariance by a convex blend

$$\Sigma_c' := (1 - \beta)\Sigma_c + \beta\Sigma_c^{\text{H}}, \quad \beta \in [0, 1]. \quad (15)$$

When $\Sigma_c^{\text{H}}$ is tighter than $\Sigma_c$ (in the Loewner order), (15) performs a principled shrinkage and thus improves Fisher separation and does not increase the optimized Cantelli surrogate along the path (see Appendix B.6). This update is lightweight, requiring only a weighted sum and a convex combination. Its simplicity can also limit flexibility: (15) may under-adapt for tail classes with multi-cluster structure or pronounced anisotropy that a single head-aggregated covariance cannot represent.

**(ii) Head-guided latent diffusion backend.** Beyond moment-matched Gaussian sampling, we also investigate a class-conditional latent DDIM (Song et al., 2020) as a complementary feature-generation backend. Since long-tailed data may bias conditional generation toward data-rich classes, especially for sparsely sampled tail classes (Zhang et al., 2024), we incorporate two stabilizing designs below.

*(a) Skeleton injection from head geometry.* Similar to (Ma et al., 2024), for each tail class $c$, we inject a head-guided "skeleton" by sampling a small set of anchor features using eigen-geometry borrowed from related heads. Concretely, using the eigen-decomposition of a head-guided covariance proxy, we draw anchors of the form $x_c^{(i)} = \mu_c + \sum_{j=1}^P \epsilon_j \lambda_j \xi_j$, $\epsilon_j \sim \mathcal{N}(0, 1)$, which expands the support of tail geometry before diffusion training. These anchors provide a structural scaffold and reduce the tendency of the generator to memorize scarce tail samples.

*(b) Mean-centering for shape-focused training.* We train diffusion on residual features $\tilde{x} = x - \mu_c$ (and anchors analogously), so the generator focuses on intra-class shape/dispersion. This accelerates convergence and decouples covariance learning from mean calibration: generated residuals are later re-centered by the calibrated mean.

Diffusion-generated tail features can yield smoother covariance estimates than raw tail statistics while preserving flexible feature variations. However, this generative backend introduces additional computational overhead. We therefore implement it with a lightweight FiLM-modulated (Perez et al., 2018) residual MLP tailored for feature generation (Appendix D).

**Tail feature synthesis.** After obtaining the calibrated mean $\mu_c'$ and calibrated covariance information, we synthesize tail features to match the maximum head-class sample count. We consider two instantiations: (i) *Gaussian sampling* assuming a simple approximation, $\hat{x} \sim \mathcal{N}(\mu_c', \Sigma_c')$, using $\Sigma_c'$ from (15); and (ii) *Diffusion sampling*, where the generator produces residuals $\hat{r}$ and we re-center by $\mu_c'$: $\hat{x} = \mu_c' + \hat{r}$. These synthesized features are then used to fine-tune the classifier in Stage-2.

**Rival-direction control for synthesized features.** Even after covariance calibration, synthesized features can still exhibit excessive dispersion along rival-induced directions, since our calibration uses a proxy $Q$ rather than the ideal one. We therefore apply a lightweight feature-level post-processing that selectively contracts variance in a low-rank rival subspace while preserving diversity in its orthogonal complement.

*Rival subspace.* For a tail class $c$, define rival directions $\{a_{c,k}\}_{k \in \mathcal{K}_c}$, e.g., $a_{c,k} := S_{c,k}^{-1}(\mu_c - \mu_k)$, $S_{c,k} := \Sigma_c + \Sigma_k$. We form an orthonormal basis $U_c = [u_{c,1}, \ldots, u_{c,r}] \in \mathbb{R}^{d \times r}$ for their span via Gram–Schmidt, with $r \le |\mathcal{K}_c|$.

*Triggering rule.* Given synthesized features $\{\hat{x}_i\}_{i=1}^m$ for class $c$, we center them by the calibrated mean $\mu_c'$ and estimate the (weighted) rival-subspace variance:

$$\hat{V}_c := \sum_{k \in \mathcal{K}_c} \omega_{c,k} \text{Var}\left(a_{c,k}^\top(\hat{x} - \mu_c')\right). \quad (16)$$

Let $V_c^{\text{base}}$ be a reference level (e.g., from the original frozen tail feature). If $(\hat{V}_c - V_c^{\text{base}})/V_c^{\text{base}} > \tau$ for a small tolerance $\tau$, we activate the contraction; otherwise we keep $\hat{x}$ unchanged.

*Low-rank Shrinkage.* We shrink only along selected axes in $U_c$: choose a diagonal matrix $\Gamma_c = \text{diag}(\gamma_{c,1}, \ldots, \gamma_{c,r})$ with $0 \le \gamma_{c,j} \le \gamma_{\max}$, and apply the linear operator:

$$\hat{x}^{\text{new}} = \mu_c' + \left(I - U_c\Gamma_c U_c^\top\right)(\hat{x} - \mu_c'). \quad (17)$$

This update leaves the mean unchanged and contracts the variance only inside the rival subspace: for each axis $u_{c,j}$, the centered projection is scaled by $(1 - \gamma_{c,j})$, hence its variance is reduced by $(1 - \gamma_{c,j})^2$, while components orthogonal to $\text{span}(U_c)$ are preserved (see Appendix B.7). In practice, we set larger $\gamma_{c,j}$ for axes with stronger rival energy and keep $\gamma_{\max}$ small to avoid over-contraction. Implementation details are provided in Appendix C.3.

### 3.3. Global Safety via a Potential Function

A natural concern is whether updating all tail classes may lead to uncontrolled interactions. To formalize *safety* at the multi-class level, we introduce a global potential that aggregates pairwise threshold-optimized surrogates. Let

$\theta := \{(\mu_c, \Sigma_c)\}_{c \in \mathcal{C}_{\text{tail}}}$ collect tail statistics, and define:

$$\Phi(\theta) := \sum_{c \in \mathcal{C}_{\text{tail}}} \sum_{k \in \mathcal{K}_c} \omega_{c,k} \, G_{c,k}(\theta). \qquad (18)$$

**Proposition 3.1.** *(First-order global safety). Let $\theta_0$ be the current tail statistics and consider a single-round calibration update $\theta^+ = \theta_0 + \Delta\theta$ produced by the per-class mean and covariance calibration operators with step sizes $(\alpha, \beta)$. Assume $\Phi$ has a locally Lipschitz gradient around $\theta_0$. If the update direction satisfies the first-order descent condition $\langle \nabla\Phi(\theta_0), \Delta\theta \rangle \leq 0$, then $\Phi$ does not increase to first order:*

$$\Phi(\theta^+) \leq \Phi(\theta_0) + O(\|\Delta\theta\|^2) = \Phi(\theta_0) + O(\alpha^2 + \beta^2). \quad (19)$$

*Here $\|\Delta\theta\| = O(\alpha + \beta)$ since the update is linear in $(\alpha, \beta)$. Proof in Appendix B.8.*

**Why RBDR remains controllable.** Proposition 3.1 shows that global safety under frozen rivals is ensured once the induced calibration update $\Delta\theta$ forms a descent direction of $\Phi$ at $\theta_0$, i.e., $\langle \nabla\Phi(\theta_0), \Delta\theta \rangle \leq 0$. Since $\Phi$ is a weighted sum of pairwise surrogates, a convenient *sufficient route* toward this is to enforce pairwise-safe updates when constructing $\Delta\theta$ from the frozen statistics: we project the mean-shift direction onto the multi-rival supportive set and restrict covariance updates to valid shrinkage/contraction paths. Together with the surrogate–Fisher link (Theorems 2.7 and 2.11), these design choices provide a practical route toward first-order safe updates under moderate step sizes.

**Single-round RBDR pipeline.** While $\Phi$ in (18) could in principle be minimized by iterative block-coordinate updates, recomputing rivals/statistics is costly; for efficiency and limited compute, RBDR uses a single-round pipeline: (i) *mean calibration* $\mu_c \leftarrow \mu_c + \alpha\,\delta_c^\star$ for each $c \in \mathcal{C}_{\text{tail}}$; (ii) *covariance calibration* to obtain $\Sigma_c'$ (via lightweight head-guided shrinkage or diffusion-based synthesis); (iii) *feature synthesis* for each tail class using $(\mu_c, \Sigma_c')$, followed by *rival-subspace contraction* as a post-processing step; (iv) *classifier fine-tuning* on frozen features augmented with the processed synthesized tail samples. The overall workflow is illustrated in Figure 1, and the complete algorithm is provided in Appendix C.4.

# 4. Experiments

## 4.1. Experimental Setting

**Datasets and Evaluation Metrics.** We evaluate RBDR on standard long-tailed recognition benchmarks, including CIFAR10/100-LT (Krizhevsky et al., 2009), ImageNet-LT (Liu et al., 2019), and iNaturalist 2018 (Van Horn et al., 2018). We measure class imbalance by the factor $\rho = n_{\max}/n_{\min}$, where $n_{\max}$ and $n_{\min}$ denote the largest and smallest training class sizes. Both CIFAR10-LT and CIFAR100-LT use $\rho \in \{200, 100, 50, 10\}$; additional details are provided in Appendix E.1.

Following standard protocols (Gao et al., 2024a), we train on the imbalanced training set and report Top-1 accuracy on the balanced test set. For ImageNet-LT and iNaturalist 2018, we additionally report accuracies on the Many/Medium/Few splits. All results are averaged over 5 random trials with different random seeds.

**Implementation Details.** Following previous work (Du et al., 2023), for CIFAR-LT, we use ResNet-32 (He et al., 2016) as the backbone. For ImageNet-LT and iNaturalist 2018, we adopt ResNeXt-50 (Xie et al., 2017) and ResNet-50 (He et al., 2016), respectively. We follow a two-stage pipeline: (i) train the backbone for 200 epochs and then freeze it; (ii) fine-tune the classifier for 50 epochs. Appendix E.2 provides implementation details, including training settings (Table 4), the OT setup (cost/marginals), the implementation of RBDR-DDIM, rival-set construction, and hyperparameter choices.

**Baseline.** We consider the following widely used baselines: (1) Cross-entropy (CE); (2) Re-weighting methods: LDAM-DRW (Cao et al., 2019), WD (Alshammari et al., 2022), DisA (Gao et al., 2024a) and Focal-SAM (Li et al., 2025b); (3) Two-stage methods: BBN (Zhou et al., 2020), GCL (Li et al., 2022) and IP-DPP (Lin & Yuan, 2025); (4) Data augmentation methods: CMO (Park et al., 2022) and CE+OTmix (Gao et al., 2024b); (5) Contrastive learning-based methods: SBCL (Hou et al., 2023) and FeatRecon (Yi et al., 2025); (6) **Feature-space distribution reconstruction methods**: Paco+BatchFormer (Hou et al., 2022), GistNet (Liu et al., 2021), LADC (Wang et al., 2022), FDC (Ma et al., 2023), FUR (Ma et al., 2024) and H2T (Li et al., 2024).

**Our methods:** We report the default Gaussian/shrinkage-based RBDR and a diffusion-based variant, RBDR-DDIM, which replaces the covariance calibrator with a class-conditional latent diffusion model. Architecture and training/sampling details of RBDR-DDIM are deferred to Appendix D.

## 4.2. Results

**Results on CIFAR-LT.** Table 1 reports Top-1 accuracy on CIFAR100-LT under four imbalance factors; results on CIFAR10-LT are deferred to Appendix E.3 due to space limits. RBDR improves over the CE baseline across all imbalance settings, with larger gains in more imbalanced regimes where tail statistics are the least reliable (e.g., 49.02% at $\rho$=200 and 53.87% at $\rho$=100). Among *feature-space distribution reconstruction* methods, RBDR achieves the strongest performance; for example, it outperforms FUR by +2.82% at $\rho$=200 and +2.97% at $\rho$=100, suggesting that

*Table 1.* Top-1 accuracy (%) of various methods on CIFAR100-LT, ImageNet-LT and iNaturalist 2018. **Bold** indicates the best in each column; underline indicates the best among feature-space reconstruction methods.

| Method | CIFAR100-LT | | | | ImageNet-LT | | | | iNaturalist 2018 | | | |
|---|---|---|---|---|---|---|---|---|---|---|---|---|
| | 200 | 100 | 50 | 10 | Many | Medium | Few | All | Many | Medium | Few | All |
| CE | 34.80 | 38.30 | 43.90 | 55.70 | 65.90 | 37.50 | 7.70 | 44.40 | 76.10 | 65.05 | 57.20 | 64.70 |
| LDAM-DRW (Cao et al., 2019) | 38.45 | 42.04 | 46.62 | 58.71 | 58.63 | 48.95 | 30.37 | 49.96 | - | - | - | 68.15 |
| BBN (Zhou et al., 2020) | - | 42.56 | 47.02 | 59.12 | 43.30 | 45.90 | 43.70 | 44.70 | 69.40 | 70.80 | 65.30 | 66.30 |
| WD (Alshammari et al., 2022) | - | 53.55 | **57.71** | **68.67** | 62.50 | 50.40 | 41.50 | 53.90 | 71.20 | 70.40 | 69.70 | 70.20 |
| GCL (Li et al., 2022) | 44.90 | 48.70 | 53.60 | - | 62.24 | 48.62 | 52.12 | 54.51 | 66.43 | 71.66 | 72.47 | 71.47 |
| CMO (Park et al., 2022) | - | 50.00 | 53.00 | 60.20 | 67.00 | 42.30 | 20.50 | 49.10 | **76.90** | 69.30 | 66.60 | 68.90 |
| SBCL (Hou et al., 2023) | - | 44.90 | 48.70 | 57.90 | 63.80 | 51.30 | 31.20 | 53.40 | 73.30 | 71.90 | 68.60 | 70.80 |
| DisA (Gao et al., 2024a) | 45.20 | 49.80 | 54.40 | 65.90 | 65.00 | 52.10 | 33.00 | 54.50 | - | - | - | - |
| CE+OTmix (Gao et al., 2024b) | - | 46.40 | 40.70 | 61.60 | **70.00** | 45.90 | 22.30 | 52.00 | 69.30 | 70.50 | 68.40 | 69.50 |
| LA+Focal-SAM (Li et al., 2025b) | 46.00 | 52.40 | 54.50 | 63.80 | 63.90 | 52.20 | 34.40 | 54.30 | 68.40 | 72.00 | 72.50 | 71.80 |
| IP-DPP (Lin & Yuan, 2025) | 46.70 | 52.40 | - | - | 59.70 | 50.80 | 32.40 | 51.70 | 72.70 | 72.90 | **75.70** | **74.00** |
| FeatRecon (Yi et al., 2025) | - | 53.41 | 57.48 | 65.67 | 68.10 | **55.30** | 38.30 | **57.80** | 72.00 | **73.90** | 73.90 | 73.70 |
| *Feature-space distribution reconstruction methods* | | | | | | | | | | | | |
| GistNet (Liu et al., 2021) | - | - | - | - | 52.80 | 39.80 | 21.70 | 42.20 | - | - | - | 70.80 |
| Paco+BatchFormer (Hou et al., 2022) | 47.80 | 52.40 | - | - | - | - | - | - | - | - | - | - |
| LADC (Wang et al., 2022) | 46.67 | 50.79 | 54.93 | 64.68 | - | - | - | 52.60 | - | - | - | 69.33 |
| FDC (Ma et al., 2023) | 45.80 | 50.60 | 54.10 | 61.30 | 65.50 | 51.90 | 37.80 | 55.30 | 72.40 | 72.60 | 72.70 | 72.20 |
| FUR (Ma et al., 2024) | 46.20 | 50.90 | 54.10 | 61.80 | 65.40 | 52.20 | 37.80 | 55.50 | 73.60 | 72.90 | 73.10 | 72.60 |
| MisLAS+H2T (Li et al., 2024) | 43.84 | 47.62 | 52.73 | - | 62.42 | 51.07 | 35.36 | 52.90 | 69.68 | 72.49 | 72.15 | 72.05 |
| GCL+H2T (Li et al., 2024) | 45.24 | 48.88 | 53.76 | - | 62.36 | 48.75 | 52.15 | 54.62 | 67.74 | 71.92 | 72.22 | 71.62 |
| **RBDR-DDIM (ours)** | 48.57 | 53.45 | 56.65 | 65.95 | 65.15 | 52.58 | 37.55 | 55.32 | 72.95 | 72.70 | 72.75 | 72.35 |
| **RBDR (ours)** | **49.02** | **53.87** | 57.12 | 66.36 | 65.72 | 53.35 | 38.25 | 55.91 | 73.64 | 73.17 | 73.29 | 72.92 |

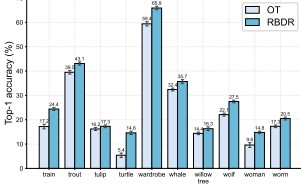

*(a)* Per-class Top-1 accuracy.

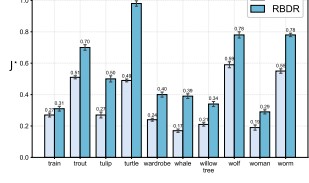

*(b)* Dim.-normalized $\overline{J^\star}$ to top-10 rivals.

*Figure 2.* Rival-oriented diagnostics on the 10 most-scarce CIFAR100-LT classes. For each tail class, we report (a) the per-class accuracy change from OT initialization to RBDR, and (b) the change of the dimension-normalized optimal Fisher separation $J^\star$ averaged over its top-10 rivals.

converting head-to-tail reconstruction cues into *bounded* statistic updates is particularly effective under severe sample scarcity.

**Results on ImageNet-LT and iNaturalist 2018.** On the two large-scale benchmarks, RBDR achieves competitive overall accuracy while consistently improving over reconstruction-based baselines. On ImageNet-LT, RBDR outperforms FD-C/FUR on Medium and Few and also improves overall accuracy. We observe that some transfer-based variants can boost Few accuracy more aggressively but at the expense of head performance: for example, GCL+H2T reaches a high Few score (52.15%) yet markedly reduces Many accuracy (62.36%), leading to a lower overall result. In contrast, RBDR is designed to be conservative and controllable: the rival-aware constraints specifically mitigate tail confusion while limiting drift along rival-aligned directions. On iNaturalist 2018, RBDR similarly yields consistent gains within the reconstruction family.

*Table 2.* Sensitivity to the number of rivals $M$ in rival-oriented diagnostics on the 10 most-scarce CIFAR100-LT classes. Results are averaged over the 10 classes and reported as OT→RBDR changes ($\Delta\overline{J^\star}$, $\Delta$Acc).

| $M$ | 1 | 5 | **10** | 20 | 30 |
|---|---|---|---|---|---|
| $\Delta\overline{J^\star}$ | 0.12 | 0.18 | **0.21** | 0.22 | 0.23 |
| $\Delta$Acc (%) | 2.5 | 4.2 | **4.7** | 4.8 | 4.8 |

Across datasets, RBDR-DDIM does not outperform the default RBDR in our setup, with a larger gap on ImageNet-LT and iNaturalist 2018. A plausible explanation is that class-conditional feature diffusion is harder to train reliably with a large number of categories and severe per-class sparsity. Moreover, in the frozen feature space, moment-matched Gaussian sampling already provides a strong reconstruction baseline, while diffusion-driven diversity may increase dispersion along rival-aligned directions and partially offset the benefit of conservative calibration.

**Quantitative Evaluation.** To directly assess whether rival-aware calibration improves the *relevant* discriminative margins, we focus on the ten most-scarce classes in CIFAR100-LT ($\rho$=100) and evaluate separability against their top-$M$ most-confusable rivals. Unless otherwise stated, we use $M$=10 and compare OT initialization with the full RBDR. As shown in Figure 2a, RBDR improves the accuracy of all ten tail classes, with larger gains on the hardest ones (e.g., *turtle*: 5.4%→14.6%), yielding a +4.7% improvement on average.

Figure 2b reports the dim.-normalized $\overline{J^\star}$ averaged over

*Table 3.* Ablation of RBDR Modules over OT Initialization.

| Method | CIFAR100-LT | | | ImageNet-LT | | | iNaturalist 2018 | | |
|---|---|---|---|---|---|---|---|---|---|
| | Few | All | $\Delta h$ | Few | All | $\Delta h$ | Few | All | $\Delta h$ |
| OT | 33.79 | 52.46 | — | 34.59 | 53.23 | — | 69.66 | 70.17 | — |
| + SP | 35.81 | 53.42 | +0.11 | 37.23 | 55.08 | +1.19 | 72.37 | 72.28 | +1.35 |
| + LS | **36.48** | **53.87** | +0.29 | **38.25** | **55.91** | +1.83 | **73.29** | **72.92** | +2.01 |

the selected rivals. While this diagnostic is not enforced during optimization and averaging can in principle obscure pairwise effects, we observe consistent increases across all ten classes. Moreover, Table 2 shows that both $\Delta\overline{J^\star}$ and $\Delta$Acc improve as $M$ grows and largely saturate beyond $M\approx10$–20, suggesting that evaluating a moderate number of top rivals is sufficient to capture the main rival-facing gains. Overall, the aligned trends between $\Delta\overline{J^\star}$ and $\Delta$Acc support that risk-bounded, rival-aware updates improve tail-class separability without relying on aggressive transfer.

### 4.3. Ablation Study

Due to space constraints, additional ablations, sensitivity analyses, and computational-cost results are deferred to Appendix E.

**Module ablation over OT initialization.** Table 3 decomposes RBDR into two plug-in components on top of the OT-based initialization: (i) supportive projection (SP) for risk-bounded mean calibration, and (ii) low-rank shrinkage (LS) for suppressing dispersion along rival-aligned directions. Here $\Delta h$ denotes the Many-split accuracy gain over OT (i.e., $\Delta h := \text{Acc}_{\text{Many}} - \text{Acc}_{\text{Many}}^{\text{OT}}$). Across CIFAR100-LT ($\rho=100$), ImageNet-LT, and iNaturalist 2018, SP already yields clear gains on the Few split while keeping $\Delta h \geq 0$. Adding LS further improves Few and All and typically increases $\Delta h$ across datasets, suggesting that rival-subspace variance control complements the conservative mean update and helps preserve head performance. Notably, the improvement is more pronounced on ImageNet-LT and iNaturalist 2018, which have larger and more fine-grained label spaces where errors concentrate among similar categories. In this regime, rival-aware constraints help mitigate drift toward frequent confusable head classes during head-guided transfer, resulting in more stable and effective reconstruction.

## 5. Conclusion

We propose *Risk-Bounded Distribution Reconstruction* (RBDR), an offline statistic calibration framework for long-tailed recognition in a standard two-stage setting. Motivated by rival-dominated tail errors, RBDR leverages a distribution-free Cantelli surrogate and its monotone connection to Fisher separation to derive directional constraints for safe statistic updates. Concretely, RBDR performs rival-aware mean calibration via supportive projection and con-

trols dispersion through covariance regularization with a lightweight rival-subspace contraction for synthesized features. Experiments on CIFAR-LT, ImageNet-LT, and iNaturalist 2018 show consistent gains over feature-space reconstruction baselines. Future work will explore how the risk-bounded calibration principle can be extended beyond offline frozen-feature reconstruction toward lightweight online long-tailed training.

## Acknowledgements

This work was supported in part by the National Natural Science Foundation of China under Grant No. 62576266, in part by the Fundamental Research Funds for the Central Universities under Grant Nos. QTZX24003 and QTZX23018, and in part by the 111 Project under Grant No. B18039.

The work of Wenchao Chen was supported in part by the National Natural Science Foundation of China under Grant No. 62571396, in part by the Fundamental Research Funds for the Central Universities under Grant No. QTZX26120, in part by the National Radar Signal Processing Laboratory under Grant No. KGJ202401, and in part by the National Key Laboratory of Electromagnetic Space Security under Grant No. JS20260300296.

## Impact Statement

This paper proposes a rival-aware, risk-bounded offline calibration framework for long-tailed recognition. The intended positive impact is to improve recognition robustness under severe class imbalance, particularly for rare categories that are often underrepresented in training data.

The method uses learned feature statistics and does not introduce new data sources, but it may still inherit dataset biases or perform unevenly under distribution shift. In sensitive applications, improved tail accuracy should therefore be accompanied by domain-specific validation, monitoring, and assessment of fairness and failure modes before deployment.

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

## A. Related Work

### A.1. Long-Tailed Learning

Long-tailed learning aims to mitigate performance degradation under severe class imbalance. Existing methods can be broadly grouped into several categories. **Re-sampling** methods rebalance the class distribution by over-sampling tail classes or under-sampling head classes. (Ren et al., 2020; Wang et al., 2020a; Zhang et al., 2021; Shi et al., 2023; Gao et al., 2024b) **Re-weighting** methods modify the training objective by assigning larger weights to tail classes or hard samples, including class-balanced losses, focal-style objectives, and margin- or logit-adjustment strategies (Lin et al., 2017; Cui et al., 2019; Cao et al., 2019; Menon et al., 2020; Guo et al., 2022a; Gao et al., 2024a; Li et al., 2025b; Sun et al., 2025). **Contrastive learning** methods improve representation geometry by explicitly encouraging class-aware separation under imbalance (Cui et al., 2021; 2023; Du et al., 2024). **Multi-expert** methods learn multiple complementary classifiers or experts to handle diverse class distributions and reduce the bias toward head classes (Zhang et al., 2022b; Yang et al., 2024).

Beyond one-stage training, **two-stage** pipelines have become a widely adopted paradigm (Kang et al., 2019; Wang et al., 2020b; Zhou et al., 2020; Li et al., 2021; Zhong et al., 2021; Li et al., 2022; Lin & Yuan, 2025). They typically learn representations in the first stage and then refine the classifier in the second stage, often combined with balanced sampling, class-wise statistic calibration, contrastive learning, or multi-objective optimization. In this work, RBDR follows the two-stage setting and performs *offline* tail distribution reconstruction on frozen features. Different from prior heuristic calibration rules, we derive risk-aware constraints to make statistic updates safer under high uncertainty in tail-statistic estimation.

### A.2. Feature-Space Distribution Reconstruction

External knowledge transfer leverages auxiliary information beyond scarce tail samples to compensate for partially observed tail distributions, and has emerged as a promising direction for long-tailed recognition. Prior work can be roughly grouped into two categories. **Semantic reconstruction** adapts strong pre-trained representations (e.g., large-scale backbones or foundation models) to long-tailed targets via fine-tuning, prompt learning, or distillation (Zhang et al., 2022a; Zhou et al., 2022; Dong et al., 2022). These approaches often improve representation quality, but usually do not provide explicit, class-wise control over tail distribution reconstruction in feature space. **Statistic-based reconstruction** instead directly modifies or synthesizes tail features by leveraging head-class statistics or geometry (Yang et al., 2021; Guo et al., 2022b; Wang et al., 2022; Hou et al., 2022; Chen & Su, 2023; Wang et al., 2023; Ma et al., 2023; 2024; 2025; Li et al., 2024; Yi et al., 2025; Li et al., 2025a). Representative strategies include calibrating tail moments using head statistics (e.g., means and covariances) (Wang et al., 2022; Ma et al., 2023; Chen & Su, 2023), head-to-tail feature blending/matching for augmentation (Hou et al., 2022; Li et al., 2024; 2025a), and integrating reconstruction with contrastive objectives to enhance discriminability (Yi et al., 2025). While effective in practice, many reconstruction rules are heuristic and may be brittle when tail statistics are highly uncertain under extreme imbalance.

**Our position.** RBDR is compatible with existing reconstruction cues (e.g., head-to-tail matching that proposes candidate update directions), but emphasizes *safe* and *controllable* offline statistic calibration. By imposing rival-aware risk constraints on mean and covariance updates, RBDR turns reconstruction signals into bounded calibration steps that better preserve multi-class separability.

## B. Proofs

### B.1. Proof of Lemma 2.4

*Proof.* Apply (4) to $Z \mid y = c$ with $a = d_c$:

$$\mathbb{P}(Z < t \mid y = c) = \mathbb{P}\big(Z - m_c(w) \leq -d_c \mid y = c\big) \leq \frac{s_c^2(w)}{s_c^2(w) + d_c^2}.$$

Similarly, apply (4) to $Z \mid y = k$ with $a = d_k$:

$$\mathbb{P}(Z \geq t \mid y = k) = \mathbb{P}\big(Z - m_k(w) \geq d_k \mid y = k\big) \leq \frac{s_k^2(w)}{s_k^2(w) + d_k^2}.$$

Substitute the two bounds into (3) to obtain (5). □

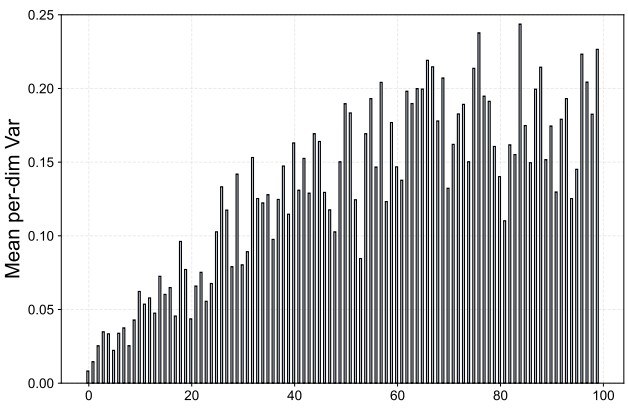

*(a)* Mean within-class variance across dimensions.

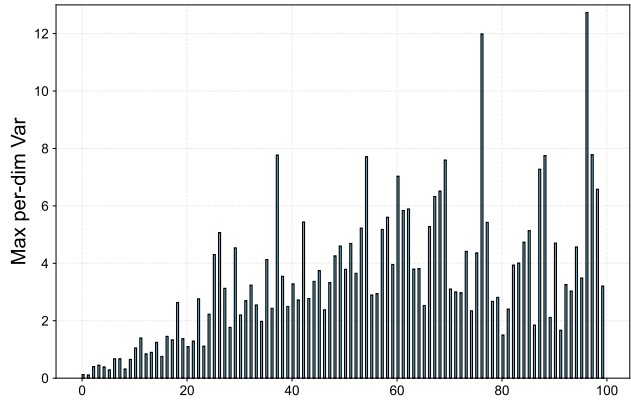

*(b)* Max within-class variance across dimensions.

*Figure 3.* Mean and maximum within-class variance across feature dimensions on CIFAR100-LT. For each class, we compute the per-dimension within-class feature variance and then aggregate it by taking the mean and the maximum across dimensions. We plot the results over classes sorted by decreasing training sample size. As class frequency decreases, variance estimates become less reliable and increasingly dispersed.

### B.2. Proof of Theorem 2.7

*Proof.* **(i)** Fix the projected variances $s_c^2 := s_c^2(w)$ and $s_k^2 := s_k^2(w)$, and consider $F(\Delta, s_c^2, s_k^2, u)$ in (6). Recall that Cantelli is applied with offsets $d_c = \frac{\Delta}{2} - u$ and $d_k = \frac{\Delta}{2} + u$, which requires $d_c \geq 0$ and $d_k \geq 0$ (i.e., the threshold lies between the two projected means); equivalently, $|u| \leq |\Delta|/2$. For any fixed $u$ satisfying $|u| \leq |\Delta_1|/2$, both $|d_c|$ and $|d_k|$ are non-decreasing in $|\Delta|$ for all $|\Delta| \geq |\Delta_1|$, and since $x \mapsto \frac{s^2}{s^2 + x^2}$ is non-increasing in $|x|$, each term in $F(\Delta, s_c^2, s_k^2, u)$ is non-increasing in $|\Delta|$ over $|\Delta| \geq |\Delta_1|$. Hence $F(\Delta, s_c^2, s_k^2, u)$ is non-increasing in $|\Delta|$ on this range.

Now take $|\Delta_2| > |\Delta_1|$ and let

$$u_1 \in \arg\min_u F(\Delta_1, s_c^2, s_k^2, u), \quad \text{so that} \quad G(\Delta_1, s_c^2, s_k^2) = F(\Delta_1, s_c^2, s_k^2, u_1).$$

By optimality, $u_1$ satisfies $|u_1| \leq |\Delta_1|/2$, and thus the monotonicity above applies to compare $\Delta_2$ with $\Delta_1$:

$$G(\Delta_2, s_c^2, s_k^2) = \min_u F(\Delta_2, s_c^2, s_k^2, u) \leq F(\Delta_2, s_c^2, s_k^2, u_1) \leq F(\Delta_1, s_c^2, s_k^2, u_1) = G(\Delta_1, s_c^2, s_k^2).$$

Therefore, $G(\Delta, s_c^2, s_k^2)$ is non-increasing in $|\Delta|$.

**(ii)** Note that the Fisher ratio is scale-invariant: for any $\gamma > 0$,

$$J(\gamma w) = \frac{(\gamma \Delta(w))^2}{(\gamma w)^\top S_w (\gamma w)} = \frac{\gamma^2 \Delta(w)^2}{\gamma^2 w^\top S_w w} = J(w).$$

Hence we may fix the scale of $w$ without loss of generality, e.g., by imposing the normalization $w^\top S_w w = 1$. Under this constraint,

$$J(w) = \frac{\Delta(w)^2}{w^\top S_w w} = \Delta(w)^2,$$

so maximizing $J(w)$ is equivalent to maximizing $|\Delta(w)|$. Combining with (i) yields that maximizing $J(w)$ favors smaller values of $G$.

Moreover, when $S_w \succ 0$, maximizing $J(w)$ is a Rayleigh-quotient problem whose optimizer satisfies

$$w^\star \propto S_w^{-1}(\mu_c - \mu_k) = (\Sigma_c + \Sigma_k)^{-1}(\mu_c - \mu_k),$$

which is the classical Fisher/LDA direction, and the optimal value equals

$$J^\star = (\mu_c - \mu_k)^\top S_w^{-1}(\mu_c - \mu_k).$$

**Clarification (pointwise vs. across-$w$).** Part (i) establishes a *pointwise* monotonicity statement: for fixed projected variances $(s_c^2, s_k^2)$, the threshold-optimized surrogate $G(\Delta, s_c^2, s_k^2)$ is non-increasing in $|\Delta|$. When varying $w$, $(s_c^2(w), s_k^2(w))$ may also change (even under the normalization $w^\top(\Sigma_c + \Sigma_k)w = 1$), so a strictly monotone relation $J(w) \uparrow \Rightarrow G(w) \downarrow$ does not hold unconditionally. Nevertheless, maximizing $J(w)$ admits two conservative connections to reducing Cantelli-style risk: (i) when projected variances are near-balanced (e.g., $s_c^2 \approx s_k^2$), the optimal threshold shift typically satisfies $u^*$ close to $0$, so the pointwise monotonic decrease in $|\Delta|$ dominates; and (ii) without any additional assumptions, the midpoint-threshold bound $\widehat{G}(\Delta) := F(\Delta, s_c^2, s_k^2, 0)$ upper-bounds $G$ pointwise and strictly decreases with $D = (\Delta/2)^2$ (hence with $J' = \Delta^2$), yielding a certified monotone diagnostic envelope (Lemma B.1) that we use only for diagnosis.

**Midpoint-threshold envelope (used for diagnosis only).** Recall $\widehat{G}(\Delta) := F(\Delta, s_c^2, s_k^2, 0)$ with $D := (\Delta/2)^2$.

**Lemma B.1** (Upper envelope and monotonicity of $\widehat{G}$). *For any $(\Delta, s_c^2, s_k^2)$ with $s_c^2, s_k^2 \geq 0$, we have*

$$G(\Delta, s_c^2, s_k^2) \leq \widehat{G}(\Delta),$$

*and $\widehat{G}$ is strictly decreasing in $D$ (hence in $J' = \Delta^2$).*

*Proof.* By definition,

$$G(\Delta, s_c^2, s_k^2) = \min_{|u| \leq |\Delta|/2} F(\Delta, s_c^2, s_k^2, u).$$

Since $u = 0$ is feasible whenever $|u| \leq |\Delta|/2$, we obtain $G(\Delta, s_c^2, s_k^2) \leq F(\Delta, s_c^2, s_k^2, 0) = \widehat{G}(\Delta)$.

Writing $\widehat{G}$ as a function of $D = (\Delta/2)^2$ gives

$$\widehat{G}(D) = \pi_c \frac{s_c^2}{s_c^2 + D} + \pi_k \frac{s_k^2}{s_k^2 + D}.$$

Differentiating yields

$$\frac{d\widehat{G}}{dD} = -\pi_c \frac{s_c^2}{(s_c^2 + D)^2} - \pi_k \frac{s_k^2}{(s_k^2 + D)^2} < 0,$$

so $\widehat{G}$ strictly decreases with $D$, equivalently with $J' = \Delta^2$. $\square$

**Remark.** We use $\widehat{G}$ only as a conservative diagnostic envelope: it upper-bounds the threshold-optimized surrogate $G$ pointwise and is provably monotone in the projected gap, without requiring additional assumptions. $\square$

### B.3. Proof of Lemma 2.8

*Proof.* Let $A := S_w^{-1} \succ 0$ and $d(\alpha) := d + \alpha\delta$. Then:

$$J^\star(\alpha) = d(\alpha)^\top A\, d(\alpha).$$

Differentiating gives

$$\frac{d}{d\alpha} J^\star(\alpha) = 2\,\delta^\top A\, d(\alpha) = 2\,\delta^\top A d + 2\alpha\,\delta^\top A\delta.$$

Since $A \succ 0$, we have $\delta^\top A\delta \geq 0$. Under the condition $\delta^\top A d > 0$, it follows that $\frac{d}{d\alpha} J^\star(\alpha) > 0$ for all $\alpha \geq 0$. Therefore, $J^\star(\alpha)$ is strictly increasing in $\alpha$ on $[0, \infty)$. $\square$

### B.4. Proof of Lemma 2.9

*Proof.* Let $S_w(\beta) = S_w - \beta Q$ with $S_w \succ 0$ and $Q \succeq 0$. Note that

$$S_w(\beta) \succ 0 \iff I - \beta\, S_w^{-1/2} Q S_w^{-1/2} \succ 0 \iff \beta < \beta_{\max} := \frac{1}{\lambda_{\max}(S_w^{-1/2} Q S_w^{-1/2})}.$$

Thus $S_w(\beta)^{-1}$ exists for all $\beta \in [0, \beta_{\max})$.

Using $\frac{d}{d\beta}A(\beta)^{-1} = -A(\beta)^{-1}A'(\beta)A(\beta)^{-1}$ and $S_w'(\beta) = -Q$, we obtain

$$\frac{d}{d\beta}S_w(\beta)^{-1} = S_w(\beta)^{-1}QS_w(\beta)^{-1} \succeq 0.$$

Therefore, for $\beta \in [0, \beta_{\max})$,

$$\frac{d}{d\beta}J^\star(\beta) = \frac{d}{d\beta}\big(d^\top S_w(\beta)^{-1}d\big) = d^\top S_w(\beta)^{-1}QS_w(\beta)^{-1}d \geq 0,$$

which proves that $J^\star(\beta)$ is non-decreasing on $[0, \beta_{\max})$. $\qquad\square$

### B.5. Proof of Proposition 2.11

*Proof.* We use a unified calibration parameter $\tau \in \mathcal{I} \subseteq \mathbb{R}$ to denote the calibration path (e.g., $\tau = \alpha$ for mean shifts or $\tau = \beta$ for covariance updates). For each rival $k \in \mathcal{K}_c$, write

$$J_{c,k}^\star(\tau) := \max_w J_{c,k}(w; \tau), \qquad G_{c,k}(\tau) := G_{c,k}(\tau),$$

and define the aggregated quantities (Definition 2.10)

$$\bar{J}_c^\star(\tau) = \sum_{k \in \mathcal{K}_c} \omega_{c,k} J_{c,k}^\star(\tau), \qquad \bar{G}_c(\tau) = \sum_{k \in \mathcal{K}_c} \omega_{c,k} G_{c,k}(\tau),$$

where $\omega_{c,k} \geq 0$ and $\sum_{k \in \mathcal{K}_c} \omega_{c,k} = 1$.

**Weighted-sum monotonicity.** Take any $\tau_2 > \tau_1$ in the admissible range. If $J_{c,k}^\star(\tau)$ is non-decreasing in $\tau$ for every $k$, then $J_{c,k}^\star(\tau_2) - J_{c,k}^\star(\tau_1) \geq 0$ for all $k$, and hence

$$\bar{J}_c^\star(\tau_2) - \bar{J}_c^\star(\tau_1) = \sum_{k \in \mathcal{K}_c} \omega_{c,k}\big(J_{c,k}^\star(\tau_2) - J_{c,k}^\star(\tau_1)\big) \geq 0,$$

so $\bar{J}_c^\star(\tau)$ is non-decreasing in $\tau$.

Moreover, for each fixed pair $(c, k)$, the surrogate–Fisher link (Theorem 2.7) implies that a non-decrease of $J_{c,k}^\star(\tau)$ along the path yields a non-increase of $G_{c,k}(\tau)$. Applying the same weighted-sum argument gives that $\bar{G}_c(\tau)$ is non-increasing.

**Instantiation by mean calibration.** Consider $\tau = \alpha$ and $\mu_c(\alpha) = \mu_c + \alpha\delta$ with $\alpha \geq 0$. For each $k \in \mathcal{K}_c$, let $d_k := \mu_c - \mu_k$ and $S_{c,k} := \Sigma_c + \Sigma_k$. If $\delta^\top S_{c,k}^{-1}d_k > 0$ holds for all $k \in \mathcal{K}_c$, then by Lemma 2.8 each $J_{c,k}^\star(\alpha)$ is non-decreasing (indeed strictly increasing) in $\alpha$ on $[0, \infty)$, and the above argument yields the monotonicity of $(\bar{J}_c^\star(\alpha), \bar{G}_c(\alpha))$.

**Instantiation by covariance calibration.** Consider $\tau = \beta$ and $\Sigma_c(\beta) = \Sigma_c - \beta Q$ with $Q \succeq 0$. For each $k \in \mathcal{K}_c$, define $S_{c,k}(\beta) := (\Sigma_c - \beta Q) + \Sigma_k$ and let $\beta_{\max}^{(k)}$ be the maximal admissible value such that $S_{c,k}(\beta) \succ 0$ for all $\beta \in [0, \beta_{\max}^{(k)})$ (as in Lemma 2.9). Let $\bar{\beta}_{\max} := \min_{k \in \mathcal{K}_c} \beta_{\max}^{(k)}$. Then for any $\beta \in [0, \bar{\beta}_{\max})$, $S_{c,k}(\beta) \succ 0$ holds for every $k$, and Lemma 2.9 implies that each $J_{c,k}^\star(\beta)$ is non-decreasing in $\beta$. The above argument again yields that $(\bar{J}_c^\star(\beta), \bar{G}_c(\beta))$ is monotone along the shrinkage path. $\qquad\square$

### B.6. Proof of Covariance Blending as Shrinkage

*Proof.* Fix a pair $(c, k)$ and let $S_{c,k}(\beta) := \Sigma_c(\beta) + \Sigma_k$. Under (15), we have

$$\Sigma_c(\beta) = (1 - \beta)\Sigma_c + \beta\Sigma_c^{\mathrm{H}} = \Sigma_c - \beta\big(\Sigma_c - \Sigma_c^{\mathrm{H}}\big).$$

Assume $\Sigma_c - \Sigma_c^{\mathrm{H}} \succeq 0$ (i.e., $\Sigma_c^{\mathrm{H}} \preceq \Sigma_c$), and define $Q := \Sigma_c - \Sigma_c^{\mathrm{H}} \succeq 0$. Then $\Sigma_c(\beta) = \Sigma_c - \beta Q$ is exactly the covariance-shrinkage path in Lemma 2.9, so $J_{c,k}^\star(\beta)$ is non-decreasing in $\beta$ over the range where $S_{c,k}(\beta) \succ 0$. By Theorem 2.7, this implies the corresponding non-increase of the threshold-optimized surrogate $G_{c,k}$. $\qquad\square$

### B.7. Proof of Covariance Change under Rival-Subspace Contraction

*Proof.* Let $\hat{x}$ denote a synthesized feature for class $c$ with mean $\mu'_c$ and covariance $\Sigma_c^{\text{gen}} := \text{Cov}(\hat{x})$. Consider the post-processing in (17):

$$\hat{x}^{\text{new}} = \mu'_c + S_c(\hat{x} - \mu'_c), \qquad S_c := I - U_c\Gamma_c U_c^\top.$$

**Mean preservation.** Since $\mathbb{E}[\hat{x} - \mu'_c] = 0$, we have

$$\mathbb{E}[\hat{x}^{\text{new}}] = \mu'_c + S_c\,\mathbb{E}[\hat{x} - \mu'_c] = \mu'_c.$$

**Covariance transformation.** Using $\text{Cov}(By) = B\,\text{Cov}(y)\,B^\top$ for any random vector $y$ with finite second moments,

$$\Sigma_c^{\text{new}} := \text{Cov}(\hat{x}^{\text{new}}) = \text{Cov}\big(S_c(\hat{x} - \mu'_c)\big) = S_c\,\Sigma_c^{\text{gen}}\,S_c^\top.$$

**Variance reduction along each axis.** Assume $U_c$ has orthonormal columns ($U_c^\top U_c = I$) and $\Gamma_c = \text{diag}(\gamma_{c,1}, \ldots, \gamma_{c,r})$. For each basis vector $u_{c,j}$ (the $j$-th column of $U_c$), we have

$$S_c u_{c,j} = \big(I - U_c\Gamma_c U_c^\top\big)u_{c,j} = u_{c,j} - U_c\Gamma_c(U_c^\top u_{c,j}) = u_{c,j} - U_c\Gamma_c e_j = (1 - \gamma_{c,j})u_{c,j}.$$

Since $S_c$ is symmetric, $u_{c,j}^\top S_c = (1 - \gamma_{c,j})u_{c,j}^\top$ as well. Therefore, the variance of the centered projection along $u_{c,j}$ satisfies

$$\text{Var}\big(u_{c,j}^\top(\hat{x}^{\text{new}} - \mu'_c)\big) = u_{c,j}^\top \Sigma_c^{\text{new}} u_{c,j} = u_{c,j}^\top S_c \Sigma_c^{\text{gen}} S_c^\top u_{c,j} = (1 - \gamma_{c,j})^2\, u_{c,j}^\top \Sigma_c^{\text{gen}} u_{c,j}.$$

Moreover, for any $v$ with $U_c^\top v = 0$, we have $S_c v = v$ and hence $v^\top \Sigma_c^{\text{new}} v = v^\top \Sigma_c^{\text{gen}} v$, i.e., variances along directions orthogonal to $\text{span}(U_c)$ are preserved. $\qquad\square$

### B.8. Proof of Proposition 3.1

*Proof.* Let $\theta(\eta) := \theta_0 + \eta\,\Delta\theta$ for $\eta \in [0, 1]$. Since $\Phi$ admits a second-order Taylor remainder at $\theta_0$, we have

$$\Phi(\theta(1)) = \Phi(\theta_0) + \frac{d}{d\eta}\Phi(\theta(\eta))\bigg|_{\eta=0} + O(\|\Delta\theta\|^2).$$

By the chain rule,

$$\frac{d}{d\eta}\Phi(\theta(\eta))\bigg|_{\eta=0} = \langle\nabla\Phi(\theta_0),\,\Delta\theta\rangle.$$

Under the assumed first-order descent condition $\langle\nabla\Phi(\theta_0),\,\Delta\theta\rangle \leq 0$, plugging into (B.8) gives

$$\Phi(\theta^+) = \Phi(\theta_0 + \Delta\theta) \leq \Phi(\theta_0) + O(\|\Delta\theta\|^2).$$

Finally, since $\Delta\theta$ is induced by mean/covariance updates with step sizes $(\alpha, \beta)$, we have $\|\Delta\theta\| = O(\alpha + \beta)$, hence $O(\|\Delta\theta\|^2) = O(\alpha^2 + \beta^2)$. $\qquad\square$

## C. Algorithm

### C.1. Hard Projection to Supportive Directions

**Supportive constraint set.** For each tail class $n \in \{1, \ldots, N\}$, we collect $K$ rival-induced constraint normals $\{v_{n,k}\}_{k=1}^K \subset \mathbb{R}^d$ and define the *supportive set*

$$\mathcal{S}_n := \big\{\delta \in \mathbb{R}^d \,:\, v_{n,k}^\top \delta \geq 0,\ \forall k = 1, \ldots, K\big\}. \tag{20}$$

Given an unconstrained candidate direction $\delta_n^{(0)}$ (e.g., an OT-driven update), our goal is to suppress rival-aligned components by projecting (hard or soft) onto $\mathcal{S}_n$.

**Hard projection via greedy half-space corrections.** We project each $\delta_n^{(0)}$ onto the polyhedral cone $\mathcal{S}_n$ by solving the Euclidean projection problem

$$\delta_n^\star \;=\; \arg\min_{\delta\in\mathbb{R}^d} \|\delta - \delta_n^{(0)}\|_2^2 \quad \text{s.t.} \quad v_{n,k}^\top \delta \geq 0, \; \forall k. \tag{21}$$

Algorithm 1 implements a greedy feasibility projection: at iteration $t$, it identifies the most violated constraint $k_n^{(t)} = \arg\min_k v_{n,k}^\top \delta_n^{(t)}$ and performs a single orthogonal correction

$$\delta_n^{(t+1)} \;=\; \delta_n^{(t)} \;-\; \frac{v_{n,k_n^{(t)}}^\top \delta_n^{(t)}}{\|v_{n,k_n^{(t)}}\|_2^2 + \epsilon}\, v_{n,k_n^{(t)}}. \tag{22}$$

When $v_{n,k_n^{(t)}}^\top \delta_n^{(t)} < 0$, the update moves $\delta_n^{(t)}$ to (approximately) satisfy $v_{n,k_n^{(t)}}^\top \delta_n^{(t+1)} \approx 0$, and the loop terminates once all constraints satisfy $v_{n,k}^\top \delta_n \geq -\tau$. This procedure is scale-invariant to $v_{n,k}$ and is highly efficient ($\mathcal{O}(NKd)$ per iteration), but with a finite budget $T$ it should be interpreted as an *approximate* Euclidean projection / feasibility projection rather than an exact nearest-point projector.

---

**Algorithm 1** Hard Projection to Supportive Directions

1: **Input:** initial directions $\delta^{(0)} \in \mathbb{R}^{N \times d}$ (rows $\delta_n^{(0)}$), constraint normals $V \in \mathbb{R}^{N \times K \times d}$ (vectors $v_{n,k}$).
2: **Hyperparameters:** iterations $T$, tolerance $\tau$.
3: **Output:** projected directions $\delta \in \mathbb{R}^{N \times d}$.

4: **Goal:** project $\delta_n^{(0)}$ onto $\mathcal{S}_n$ for all $n$ (up to tolerance).

5: Initialize $\delta \leftarrow \delta^{(0)}$.
6: **for** $t = 1$ **to** $T$ **do**
7:     Compute inner products $G_{n,k} \leftarrow v_{n,k}^\top \delta_n$ for all $(n,k)$.
8:     Find most violated constraint per row: $k_n \leftarrow \arg\min_k G_{n,k}$ and $m_n \leftarrow \min_k G_{n,k}$ for $n = 1, \ldots, N$.
9:     **if** $m_n \geq -\tau$ for all $n$ **then**
10:         **break**
11:     **end if**
12:     Gather violated normals $v_n \leftarrow v_{n,k_n}$ for $n = 1, \ldots, N$.
13:     Compute step sizes $\alpha_n \leftarrow \dfrac{m_n}{\|v_n\|_2^2 + \epsilon}$ for $n = 1, \ldots, N$.
14:     Update $\delta_n \leftarrow \delta_n - \alpha_n v_n$ for $n = 1, \ldots, N$.
15: **end for**
16: **return** $\delta$

---

### C.2. Soft Projection to Supportive Directions

As a smoother alternative, we replace the hard constraints by a hinge-squared penalty and solve the convex objective

$$\min_{\delta\in\mathbb{R}^{N\times d}} \|\delta - \delta^{(0)}\|_F^2 \;+\; \lambda \cdot \frac{1}{K} \sum_{n=1}^{N} \sum_{k=1}^{K} \Big[ -v_{n,k}^\top \delta_n \Big]_+^2, \tag{23}$$

where $\delta_n$ denotes the $n$-th row of $\delta$ and $[x]_+ = \max(x, 0)$. Algorithm 2 summarizes the resulting optimization procedure, with a PyTorch-style implementation sketch provided in Listing 1. It applies a few steps of gradient descent: letting $G_{n,k} = v_{n,k}^\top \delta_n$ and $s_{n,k} = [-G_{n,k}]_+$, the per-row gradient is

$$\nabla_{\delta_n} \;=\; 2(\delta_n - \delta_n^{(0)}) \;-\; \frac{2\lambda}{K} \sum_{k=1}^{K} s_{n,k}\, v_{n,k}, \qquad \delta_n \leftarrow \delta_n - \eta\, \nabla_{\delta_n}. \tag{24}$$

Compared to hard projection, the soft variant yields a *risk-adaptive* correction: violations with larger magnitude (more negative $v_{n,k}^\top \delta_n$) induce stronger pull-backs. It is typically more stable under large mean step sizes (large calibration $\alpha$), and can preserve useful components of $\delta_n^{(0)}$ while suppressing rival-aligned directions. Note that, unlike the hard update, the soft

---

**Algorithm 2** Soft Projection to Supportive Directions

---

1: **Input:** initial directions $\delta^{(0)} \in \mathbb{R}^{N \times d}$ (rows $\delta_n^{(0)}$), constraint normals $V \in \mathbb{R}^{N \times K \times d}$ (vectors $v_{n,k}$).

2: **Hyperparameters:** penalty $\lambda > 0$, step size $\eta > 0$, iterations $T$, optional tolerance $\tau$.

3: **Output:** refined directions $\delta \in \mathbb{R}^{N \times d}$.

4: **Objective:** $\min_\delta \|\delta - \delta^{(0)}\|_F^2 + \lambda \cdot \frac{1}{K} \sum_{n=1}^{N} \sum_{k=1}^{K} \left[ -v_{n,k}^\top \delta_n \right]_+^2$.

5: Initialize $\delta \leftarrow \delta^{(0)}$.

6: **for** $t = 1$ **to** $T$ **do**

7:     Compute inner products $G_{n,k} \leftarrow v_{n,k}^\top \delta_n$ for all $(n, k)$.

8:     Compute violations $s_{n,k} \leftarrow \left[ -G_{n,k} \right]_+ = \max(0, -G_{n,k})$.

9:     Data gradient: $\nabla_\delta^{\text{data}} \leftarrow 2(\delta - \delta^{(0)})$.

10:     Coefficients: $c_{n,k} \leftarrow \frac{2\lambda}{K} s_{n,k}$.

11:     Penalty gradient (row-wise): $\nabla_{\delta_n}^{\text{pen}} \leftarrow - \sum_{k=1}^{K} c_{n,k} \, v_{n,k}$, for $n = 1, \dots, N$.

12:     Total gradient: $\nabla_\delta \leftarrow \nabla_\delta^{\text{data}} + \nabla_\delta^{\text{pen}}$.

13:     Gradient step: $\delta \leftarrow \delta - \eta \, \nabla_\delta$.

14:     **if** $\tau$ is provided **then**

15:         $v_{\max} \leftarrow \max_{n,k} s_{n,k}$.

16:         **if** $v_{\max} < \tau$ **then**

17:             **break**

18:         **end if**

19:     **end if**

20: **end for**

21: **return** $\delta$

---

penalty is *not* scale-invariant to $\|v_{n,k}\|$; leaving $v_{n,k}$ unnormalized implicitly weights constraints by their magnitude, which can be beneficial in practice.

```python
def soft_supportive_projection(delta0, V, lam=5.0, lr=0.1, iters=2):
    """
    Softly projects the initial calibration direction onto the
    rival-supportive region.

    delta0: (N, d), initial OT directions for N tail classes
    V:      (N, K, d), rival constraint directions
            v_{c,k} = w_c - w_k

    Objective:
        min_delta ||delta - delta0||^2
        + lam * mean_k [ - <v_{c,k}, delta_c> ]_+^2
    """

    delta = delta0.clone()
    N, K, d = V.shape

    if K == 0:
        return delta

    for _ in range(iters):
        # Constraint scores: <v_{c,k}, delta_c>.
        scores = (V * delta.unsqueeze(1)).sum(dim=-1)

        # Penalize only violated constraints.
        violation = clamp(-scores, min=0.0)

        grad_data = 2.0 * (delta - delta0)
        grad_cons = -(2.0 * lam / K) * \
            (violation.unsqueeze(-1) * V).sum(dim=1)

        delta = delta - lr * (grad_data + grad_cons)

    return delta
```

*Listing 1.* Soft rival-aware supportive projection used in RBDR.

## C.3. Dangerous Low-Rank Shrinkage

After covariance calibration, synthesized features may still be over-dispersed along rival-induced directions due to using a proxy shrinkage operator. We therefore post-process synthesized samples by contracting variance only in a low-rank *dangerous* (rival) subspace while keeping the mean and the orthogonal complement unchanged.

**Rival subspace and trigger.** For tail class $c$, we collect rival directions $\{v_k\}_{k=1}^K$ (e.g., $a_{c,k} = S_{c,k}^{-1}(\mu_c - \mu_k)$ with $S_{c,k} = \Sigma_c + \Sigma_k$) and form an orthonormal basis $U \in \mathbb{R}^{d \times r}$ for their span ($r \leq r_{\max}$). Given synthesized samples $X \in \mathbb{R}^{B \times d}$ centered by $\mu_c$, let $R = X - \mathbf{1}\mu_c^\top$, $Y = RU$, and $\hat{v} = \mathrm{Var}(Y, \dim = 0) \in \mathbb{R}^r$. If a baseline variance $v^{\mathrm{base}}$ is available, we activate shrinkage only when the total dangerous energy increases beyond a tolerance, e.g., $\sum_j \hat{v}_j > \sum_j v_j^{\mathrm{base}} + \tau(\sum_j v_j^{\mathrm{base}} + \epsilon)$.

**Objective and update.** We apply a mean-preserving low-rank contraction

$$x_{\mathrm{new}} = \mu_c + (I - U\Gamma U^\top)(x - \mu_c), \qquad \Gamma = \mathrm{diag}(\gamma_1, \ldots, \gamma_r), \ 0 \leq \gamma_j \leq \gamma_{\max}. \tag{25}$$

Equivalently, this minimizes a proximal objective that penalizes hinge-excess variance in the dangerous coordinates:

$$\min_{X_{\mathrm{new}}} \|X_{\mathrm{new}} - X\|_F^2 + \sum_{j=1}^r \lambda_j \left[ \frac{\mathrm{Var}((X_{\mathrm{new}} - \mathbf{1}\mu_c^\top)u_j) - v_j^{\mathrm{base}}}{v_j^{\mathrm{base}} + \epsilon} \right]_+. \tag{26}$$

Algorithm 3 summarizes the low-rank rival-subspace contraction, and Listing 2 provides a compact implementation-oriented sketch. The algorithm sets $\gamma$ by an excess-variance rule $\gamma_j = \min(\eta\,[(\hat{v}_j - v_j^{\mathrm{base}})/(v_j^{\mathrm{base}} + \epsilon)]_+, \gamma_{\max})$ (optionally on selected axes $\mathcal{J}$), yielding the efficient batch update $R_{\mathrm{new}} = R - (Y \odot \gamma^\top)U^\top$, $X_{\mathrm{new}} = \mathbf{1}\mu_c^\top + R_{\mathrm{new}}$. This scales the $j$-th dangerous coordinate by $(1 - \gamma_j)$, equivalently its variance by $(1 - \gamma_j)^2$, while leaving the orthogonal complement unchanged, with cost $\mathcal{O}(Bdr)$ for small $r$.

```python
def lowrank_rival_shrinkage(X, center, directions, gamma=0.2, r_max=8):
    """
    Shrinks synthesized features along dangerous rival subspaces
    while preserving the class mean.

    X:          (B, d), synthesized tail features
    center:     (d,), calibrated class mean
    directions: (K, d), rival directions, e.g., w_c - w_k
    """

    if directions is None:
        return X

    # Build an orthonormal basis of the dangerous subspace.
    U = orthonormal_basis(directions, rank=r_max)

    if U is None:
        return X

    # Center features and project them onto the dangerous subspace.
    residual = X - center.unsqueeze(0)
    coeff = residual @ U

    # Low-rank shrinkage:
    # residual_new = residual - gamma * coeff @ U.T
    residual_new = residual - gamma * (coeff @ U.T)

    return center.unsqueeze(0) + residual_new
```

*Listing 2.* Low-rank shrinkage along dangerous rival subspaces.

## C.4. Workflow of RBDR

RBDR follows the *train–calibrate–fine-tune* workflow. After standard ERM training, we freeze the backbone and perform a single-round offline calibration on feature statistics: (i) *rival-aware mean calibration* for each $c \in \mathcal{C}_{\mathrm{tail}}$ via $\mu_c \leftarrow \mu_c + \alpha\,\delta_c^\star$, where $\delta_c^\star$ is obtained by projecting the OT candidate direction onto the supportive set induced by rivals; (ii) *covariance calibration* to produce $\Sigma_c'$ using either head-guided shrinkage or (optionally) a head-guided latent diffusion model; (iii)

---

**Algorithm 3** Dangerous Low-Rank Shrinkage

---

1: **Input:** samples $X \in \mathbb{R}^{B \times d}$ (rows $x_b$), class mean $\mu_c \in \mathbb{R}^d$, danger directions $V \in \mathbb{R}^{K \times d}$ (rows $v_k$).
2: **Optional inputs:** baseline variances $v^{\text{base}} \in \mathbb{R}^r$, threshold $\tau$, axis set $\mathcal{J}$, rank cap $r_{\max}$.
3: **Hyperparameters:** $\gamma_{\max} \in (0, 1)$, $\eta > 0$, $\epsilon > 0$.
4: **Output:** shrunk samples $X_{\text{new}} \in \mathbb{R}^{B \times d}$.

5: **Goal:** shrink the residual energy in the dangerous subspace while keeping the mean unchanged: $x_{\text{new}} = \mu_c + S_c(x - \mu_c)$, with $S_c$ being a low-rank contraction.

6: **if** $V$ is None **then**
7:     **return** $X$
8: **end if**
9: Construct an orthonormal basis $U \in \mathbb{R}^{d \times r}$ for $\text{span}(V)$ with rank cap $r_{\max}$.
10: **if** $U$ is None **then**
11:     **return** $X$
12: **end if**
13: Residuals: $R \leftarrow X - \mathbf{1}\mu_c^\top \in \mathbb{R}^{B \times d}$.
14: Project to dangerous subspace: $Y \leftarrow RU \in \mathbb{R}^{B \times r}$.
15: Sample variances on dangerous axes: $\hat{v} \leftarrow \text{Var}(Y, \dim = 0) \in \mathbb{R}^r$.
16: **if** $v^{\text{base}}$ is provided **and** $\tau$ is provided **then**
17:     $V \leftarrow \sum_{j=1}^r \hat{v}_j, \quad V_{\text{base}} \leftarrow \sum_{j=1}^r v_j^{\text{base}}$
18:     **if** $(V - V_{\text{base}}) \leq \tau\,(V_{\text{base}} + \epsilon)$ **then**
19:         **return** $X$
20:     **end if**
21: **end if**
22: **if** $v^{\text{base}}$ is provided **then**
23:     $q \leftarrow \max\left(0, \dfrac{\hat{v} - v^{\text{base}}}{v^{\text{base}} + \epsilon}\right)$
24:     $\gamma \leftarrow \min(\eta\,q,\ \gamma_{\max}) \in \mathbb{R}^r$
25: **else**
26:     $\gamma \leftarrow \gamma_{\max}\mathbf{1} \in \mathbb{R}^r$
27: **end if**
28: **if** $\mathcal{J}$ is provided **then**
29:     $\gamma_j \leftarrow 0$ for all $j \notin \mathcal{J}$
30: **end if**
31: Low-rank shrinkage: $R_{\text{new}} \leftarrow R - (Y \odot \gamma^\top)U^\top \in \mathbb{R}^{B \times d}$
32: Mean-preserving reconstruction: $X_{\text{new}} \leftarrow \mathbf{1}\mu_c^\top + R_{\text{new}}$
33: **return** $X_{\text{new}}$

---

*tail feature synthesis* from $(\mu_c, \Sigma_c')$, followed by an optional *rival-subspace contraction* to suppress excessive dispersion along dangerous directions; (iv) *classifier fine-tuning* on the frozen features augmented with the processed synthesized tail samples. Algorithm 4 summarizes the complete RBDR procedure, while Listing 3 gives a compact implementation-oriented sketch of its Gaussian sampling and shrinkage-based instantiation.

```python
def RBDR(class_stats, features, head_ids, tail_ids, class_counts, rival_dirs,
         alpha=0.3, beta=0.5):

    # 1. Collect class-wise means and covariances.
    mu_h = stack([class_stats[h].mean for h in head_ids])
    mu_t = stack([class_stats[t].mean for t in tail_ids])
    cov_h = [class_stats[h].cov for h in head_ids]

    # 2. Compute OT-based head-to-tail transfer.
    T = sinkhorn_transport(
        source=mu_h,
        target=mu_t,
        class_counts=class_counts,
        cost="cosine",
        eps=2.0,
        max_iter=200
    )
    T = normalize_columns(T)

    # 3. Obtain the initial mean-calibration direction.
    ot_target = T.T @ mu_h
    delta = ot_target - mu_t

    # 4. Enforce rival-aware supportive constraints.
    V = stack([rival_dirs[t] for t in tail_ids])
    delta = soft_supportive_projection(delta, V)

    # 5. Keep original head features for classifier fine-tuning.
    X_ft, y_ft = collect_features(features, head_ids)

    # 6. Reconstruct each tail-class distribution.
    for i, t in enumerate(tail_ids):
        mu_tail = class_stats[t].mean
        cov_tail = class_stats[t].cov

        # Head-guided covariance calibration.
        cov_transfer = sum(T[j, i] * cov_h[j]
                           for j in range(len(head_ids)))

        mu_new = mu_tail + alpha * delta[i]
        cov_new = (1 - beta) * cov_tail + beta * cov_transfer
        cov_new = cov_new + 1e-6 * identity_like(cov_new)

        # Gaussian feature synthesis.
        n_samples = max_num_samples(features, head_ids)
        X_syn = sample_gaussian(mu_new, cov_new, n_samples)

        # Suppress dispersion along dangerous rival subspaces.
        X_syn = lowrank_rival_shrinkage(
            X_syn,
            center=mu_new,
            directions=rival_dirs[t]
        )

        X_ft.append(X_syn)
        y_ft.append(full_label(t, n_samples))

    return X_ft, y_ft
```

*Listing 3.* Python-style pseudocode of the RBDR reconstruction pipeline.

# D. RBDR-DDIM: Latent Diffusion Backend Details

This section provides implementation details for RBDR-DDIM, an optional latent diffusion backend for feature synthesis in RBDR.

**Training features and mean centering.** We train the diffusion model in the backbone feature space. Given a feature vector $x \in \mathbb{R}^D$ with class label $y$, we first compute the empirical class mean $\mu_y$ and center the feature by $x \leftarrow x - \mu_y$

---

**Algorithm 4** Workflow of RBDR (Single-round Offline Calibration + Fine-tuning)

---

1: **Require:** Imbalanced training set $\mathcal{D}$; backbone $f_{\theta_1}$ and classifier $g_{\theta_2}$; head/tail splits $(\mathcal{C}_{\text{head}}, \mathcal{C}_{\text{tail}})$; mean step size $\alpha$
   and cov step size $\beta$; training epochs $T_{s_1}$ and fine-tuning epochs $T_{s_2}$; OT weights $(p_h, q_c)$ and cost $C$; rival sets $\{\mathcal{K}_c\}$
   and weights $\{\omega_{c,k}\}$; (optional) trigger tolerance $\tau$, contraction cap $\gamma_{\max}$, scale $\eta$.
2: **Output:** Model parameters $\{\theta_1, \theta_2\}$.

3: **Stage 1: Initial Training**
4: **for** $e = 1, 2, \ldots, T_{s_1}$ **do**
5:     Sample a mini-batch from $\mathcal{D}$ and optimize $\theta_1, \theta_2$ by minimizing cross-entropy.
6: **end for**

7: **Stage 2: Offline Calibration & Tail Feature Synthesis**
8: Freeze $\theta_1$ and extract features $x = f_{\theta_1}(\cdot)$ for all samples in $\mathcal{D}$.
9: Estimate per-class statistics $\{(\mu_c, \Sigma_c)\}_{c \in \mathcal{C}}$ and construct rival sets $\{\mathcal{K}_c\}_{c \in \mathcal{C}_{\text{tail}}}$.
10: Construct discrete measures $P = \sum_{h \in \mathcal{C}_{\text{head}}} p_h \, Dirac(\mu_h)$ and $Q = \sum_{c \in \mathcal{C}_{\text{tail}}} q_c \, Dirac(\mu_c)$.
11: Solve OT $T^{\star} \in \arg\min_{T \in \Pi(P,Q)} \langle T, C \rangle_F$ (e.g., Sinkhorn), and compute $\pi_{c,h} = \frac{T^{\star}_{h,c}}{\sum_{h'} T^{\star}_{h',c}}$.
12: **for** each tail class $c \in \mathcal{C}_{\text{tail}}$ **do**
13:     Candidate direction $\delta_c^{OT} \leftarrow \sum_{h \in \mathcal{C}_{\text{head}}} \pi_{c,h}(\mu_h - \mu_c)$                      Eq. (12)
14:     Supportive direction $\delta_c^{\star} \leftarrow \arg\min_{\delta} \|\delta - \delta_c^{OT}\|_2^2$
15:         s.t. $\delta^{\top} S_{c,k}^{-1}(\mu_c - \mu_k) \geq 0, \ \forall k \in \mathcal{K}_c$                      Eq. (13)
16:     Update mean $\mu_c \leftarrow \mu_c + \alpha \, \delta_c^{\star}$                      Eq. (14)
17:     *Covariance calibration (choose one):*
18:         **(i) Head-guided shrinkage:**
19:             $\Sigma_c^{H} \leftarrow \sum_{h \in \mathcal{C}_{\text{head}}} \pi_{c,h} \Sigma_h$
20:             $\Sigma_c' \leftarrow (1 - \beta)\Sigma_c + \beta \, \Sigma_c^{H}$                      Eq. (15)
21:         **(ii) Head-guided latent diffusion (optional):**
22:             Sample head-skeleton anchors using eigen-geometry of $\Sigma_c^{H}$.
23:             Center features by $\tilde{x} = x - \mu_c$ and train a class-conditional latent DDIM on $\tilde{x}$ (and anchors).
24:     *Tail feature synthesis (choose one):*
25:         **Gaussian sampling:** sample $\hat{x} \sim \mathcal{N}(\mu_c, \Sigma_c')$.
26:         **Diffusion sampling:** sample residual $\hat{r}$ from DDIM and set $\hat{x} = \mu_c + \hat{r}$.
27:     *Rival-direction control (optional):*
28:     Compute baseline $V_c^{\text{base}}$ from frozen tail features and estimate $\hat{V}_c$ on synthesized $\hat{x}$                      Eq. (16)
29:     **if** $(\hat{V}_c - V_c^{\text{base}})/V_c^{\text{base}} > \tau$ **then**
30:         Build rival-subspace basis $U_c$ from $\{a_{c,k}\}_{k \in \mathcal{K}_c}$ and choose $\Gamma_c = \text{diag}(\gamma_{c,1}, \ldots, \gamma_{c,r})$.
31:         Apply contraction $\hat{x} \leftarrow \mu_c + (I - U_c \Gamma_c U_c^{\top})(\hat{x} - \mu_c)$                      Eq. (17)
32:     **end if**
33:     Store processed synthesized tail features $\hat{\mathcal{Z}}_c$.
34: **end for**
35: Form augmented feature set $\mathcal{Z} \leftarrow \mathcal{Z}^{\text{head}} \cup \bigcup_{c \in \mathcal{C}_{\text{tail}}} \hat{\mathcal{Z}}_c$.

36: **Classifier Fine-tuning:**
37: **for** $e = 1, 2, \ldots, T_{s_2}$ **do**
38:     Sample a mini-batch from $\mathcal{Z}$ and fine-tune $\theta_2$ (with $\theta_1$ frozen).
39: **end for**
40: **return** $\{\theta_1, \theta_2\}$.

---

(saved for later recovery). This makes the diffusion model focus on the intra-class residual geometry rather than the absolute location.

**Forward diffusion and training objective.** We adopt a standard Gaussian diffusion process with $T$ steps and a linear $\beta$ schedule. At each iteration, we sample a timestep $t \sim \text{Unif}\{0, \ldots, T-1\}$ and draw $\epsilon \sim \mathcal{N}(0, I)$, then construct the noisy feature $x_t = \sqrt{\bar{\alpha}_t}\, x_0 + \sqrt{1 - \bar{\alpha}_t}\, \epsilon$. The noise network $\hat{\epsilon}_\theta(\cdot)$ is trained to predict $\epsilon$ with the MSE loss $\mathcal{L} = \|\hat{\epsilon}_\theta(x_t, t, y) - \epsilon\|_2^2$.

**Noise network.** To suit our feature-generation setting, we design $\hat{\epsilon}_\theta$ as a lightweight timestep- and class-conditional residual MLP. We embed $t$ with a sinusoidal positional encoding followed by a two-layer MLP, and embed the class label with a learnable embedding followed by a two-layer MLP; the conditioning vector is $\text{cond} = \text{TimeEmb}(t) + \text{ClassEmb}(y)$. The input feature is first projected to width $W$, processed by $N$ FiLM residual blocks (Figure 4), and then projected back to $D$ dimensions. Each block applies LayerNorm–Linear–SiLU, FiLM modulation $h \leftarrow h \odot (1 + \gamma) + \beta$ with $(\gamma, \beta) = \text{Linear}(\text{cond})$, followed by Linear–SiLU–Dropout and a residual skip connection.

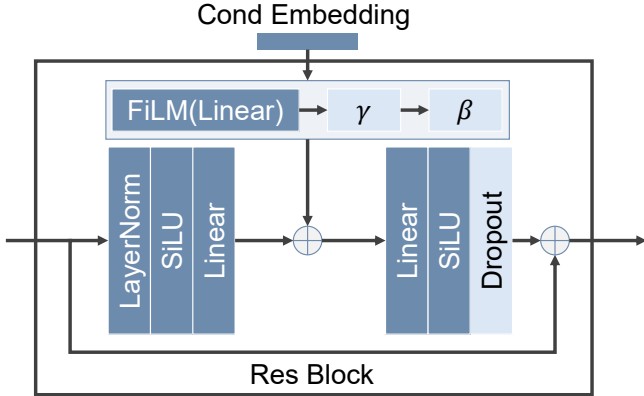

*Figure 4.* FiLM-MLP residual block.

**Classifier-free guidance (CFG).** We adopt classifier-free guidance (CFG) by reserving label 0 as the unconditional token and shifting all class labels by $+1$ during training. With probability $0.1$, we drop the class condition by replacing the label with 0 (i.e., $p_{\text{uncond}} = 0.1$). At inference, we perform deterministic DDIM sampling ($\eta = 0$) with 10 sampling steps by default, and apply CFG as $\epsilon = (1 + w)\epsilon_{\text{cond}} - w\epsilon_{\text{uncond}}$.

**DDIM sampling and feature recovery.** We generate residual features with deterministic DDIM ($\eta = 0$) using a reduced number of steps (e.g., 10). Starting from $x_T \sim \mathcal{N}(0, I)$, we iteratively update $x_t$ using DDIM with the guided $\epsilon$ above. After sampling, we recover the final class-conditional feature by adding the target class mean: $\tilde{x} = x + \mu_y$. Notably, this design allows us to plug in either the original empirical mean or the RBDR-calibrated mean, while keeping the diffusion model responsible only for residual shape modeling.

## E. More Experimental Setting

### E.1. Dataset

**Benchmark Datasets.** We conduct experiments on four widely used long-tailed datasets: CIFAR10-LT, CIFAR100-LT, ImageNet-LT, and iNaturalist. Below is a detailed description of these datasets:

- **CIFAR10-LT/CIFAR100-LT** are imbalanced versions of the CIFAR datasets, both containing 50,000 training samples and 10,000 validation samples. The difference is that CIFAR10-LT has 10 classes, while CIFAR100-LT has 100 classes. To create the long-tailed imbalanced datasets, the number of selected samples in the $c$-th category is determined as $n_c \mu^c$, where $\mu \in (0, 1)$, as is commonly done. We construct different variants of each dataset with varying imbalance factors $\rho = 200, 100, 50,$ and 10 to evaluate performance.

- **ImageNet-LT** is an imbalanced version of the ImageNet 2012 dataset, containing 1,000 classes with a total of 115,846 training samples. The largest class has 1,280 samples, while the smallest has 5, with $\rho = 256$. The validation set is

balanced, consisting of 50,000 samples.

- **iNaturalist 2018** is a large, real-world imbalanced dataset containing 8,142 classes with a total of 437,513 samples. The largest class has 1,000 samples, while the smallest has only 2, resulting in an imbalance factor of $\rho = 500$. Each class has three validation samples, forming a balanced validation set of 24,426 samples.

### E.2. Implementation Details

The overall workflow of RBDR is summarized in Appendix C.4.

**OT computation.** For OT computation, we use the Sinkhorn algorithm with entropy regularization $\varepsilon = 2$, at most 200 iterations, and early stopping when the average $\ell_1$ change of the dual variables between consecutive iterations is below $10^{-1}$. Inspired by prior work, we define the cost matrix $C$ by $C_{h,c} = 1 - \cos(\mu_h, \mu_c)$.

Note that the head/tail partition is used only for executing RBDR and is distinct from the Many/Medium/Few splits reported for evaluation. Specifically, we label a class as *head* if its training sample size exceeds half of the maximum class size, i.e., $n_c > \frac{1}{2} n_{\max}$, and treat the remaining classes as *tail* to be calibrated. This partition does not affect the evaluation protocol; it only determines which classes provide transfer statistics (heads) and which ones receive calibration (tails). To set the OT marginals, a simple choice is uniform weights: $p_h = \frac{1}{H}$ for head classes and $q_c = \frac{1}{T}$ for tail classes, where $H$ and $T$ denote the numbers of head and tail classes. In our implementation, we use sample-aware weights to reflect estimation reliability: $p_h = \frac{\exp(n_h/\tau_{\mathrm{w}})}{\sum_{h=1}^{H} \exp(n_h/\tau_{\mathrm{w}})}$ to emphasize larger head classes, and $q_c = \frac{\exp(1/(\tau_{\mathrm{w}} n_c))}{\sum_{t=1}^{T} \exp(1/(\tau_{\mathrm{w}} n_t))}$ to upweight rarer tail classes.

**Rival set construction.** For each tail class $c$, we construct its rival set $\mathcal{K}_c \subseteq [K] \backslash \{c\}$ using the frozen classifier. Concretely, we compute a class-wise logit vector for $c$ by averaging per-sample logits, i.e., $\bar{\ell}_c = \frac{1}{n_c} \sum_{i:y_i=c} \ell(x_i)$, exclude $\bar{\ell}_{c,c}$, and select the top-$M$ classes with the largest logits as $\mathcal{K}_c$ (i.e., the top-$M$ most confusable classes under the frozen classifier). We set $M = \lceil 0.1K \rceil$ (10% of the total number of classes). A sensitivity analysis of $M$ is provided in Appendix E.5.

**RBDR-DDIM implementation.** For a tail class $c$, we select its most similar head class (by cosine similarity between class means) to provide head-guided second-order statistics, and then draw 100 skeleton features via $x_c^{(i)} = \mu_c + \sum_{j=1}^{P} \epsilon_j \lambda_j \xi_j$, $\epsilon_j \sim \mathcal{N}(0, 1)$, where $\{(\lambda_j, \xi_j)\}_{j=1}^{P}$ are the eigen-components of the calibrated covariance. We then train a class-conditional DDIM for 50 epochs on these skeleton-augmented tail features to model residual variations. The DDIM noise network $\hat{\epsilon}_\theta$ is instantiated as a residual MLP with FiLM modulation, comprising 8 FiLM-MLP ResBlocks.

**Dangerous-direction approximation via classifier weights.** When computing rival-aligned (dangerous) directions, we avoid explicitly forming $\mathbf{S}_w^{-1}(\mu_c - \mu_k)$ and instead approximate it using the frozen classifier weights:

$$v_{c,k} \propto w_c - w_k. \tag{27}$$

This choice is motivated by the classical Gaussian discriminant/LDA model with shared within-class covariance $\boldsymbol{\Sigma}$, where the Bayes-optimal linear classifier satisfies $w_c = \boldsymbol{\Sigma}^{-1}\mu_c$ (up to bias terms). Therefore,

$$w_c - w_k = \boldsymbol{\Sigma}^{-1}(\mu_c - \mu_k) \propto \mathbf{S}_w^{-1}(\mu_c - \mu_k), \tag{28}$$

showing that weight differences recover the desired whitened mean-gap direction under the generative optimum. Practically, this approximation (i) avoids per-class matrix inversions and substantially reduces computation, and (ii) is more numerically robust when tail-class $\mathbf{S}_w$ is ill-conditioned or nearly singular due to scarce samples. Since $w_c$ is readily available from the frozen classifier and is typically better behaved than tail covariance estimates, $w_c - w_k$ provides a stable proxy for constructing rival-aligned constraints.

**Supportive projection.** For supportive-direction computation, we use the *soft* variant by default (Appendix C.2) with fixed hyperparameters: penalty coefficient $\lambda = 5$, learning rate $\mathrm{lr} = 0.1$, and iterations $\mathrm{iter} = 2$.

**Low-rank shrinkage.** For the low-rank shrinkage module (Appendix C.3), we fix $\tau = 0.05$, rank cap $r_{\max} = 8$, and $\gamma_{\max} = 0.2$.

**Other settings.** Following prior work (Du et al., 2023; Sun et al., 2025), we adopt standard image augmentations during backbone training, including random cropping, color jittering, random grayscale, Gaussian blur, and random horizontal flipping. We set $\alpha = 0.3$ on CIFAR-10/100-LT and $\alpha = 0.1$ on ImageNet-LT and iNaturalist 2018, and use $\beta = 0.5$ by default. The effects of $\alpha$ and $\beta$ are analyzed in Appendix E.4. Table 4 summarizes the training settings for all datasets. All experiments are conducted on two NVIDIA RTX 4090 GPUs.

*Table 4.* Implementation details on four benchmark datasets

| Dataset | Backbone | SGD | | | Learning rate (LR) | | LR decay | Warm-up | Batch size | Epoch | |
| --- | --- | --- | --- | --- | --- | --- | --- | --- | --- | --- | --- |
| | | Weight decay | Momentum | | Stage 1 | Stage 2 | | | | Stage 1 | Stage 2 |
| CIFAR10-LT | ResNet-32 | 5e-3 | | | 5e-2 | 1e-2 | | ✗ | 128 | 200 | 50 |
| CIFAR100-LT | ResNet-32 | 5e-3 | 0.9 | | 5e-2 | 1e-2 | Cosine | ✗ | 128 | 200 | 50 |
| ImageNet-LT | ResNeXt-50 | 2e-4 | | | 1e-1 | 1e-4 | annealing | ✓ | 256 | 200 | 50 |
| iNaturalist 2018 | ResNet-50 | 2e-4 | | | 1e-1 | 1e-4 | | ✓ | 256 | 200 | 50 |

## E.3. Results on CIFAR10-LT

**Results on CIFAR10-LT.** Table 5 reports results on CIFAR10-LT. RBDR yields consistent improvements over CE across imbalance factors, reaching 81.69% ($\rho$=200), 85.57% ($\rho$=100), and 88.52% ($\rho$=50). Within *feature-space distribution reconstruction* methods, RBDR attains the best or near-best accuracy (e.g., +1.89% over FUR at $\rho$=200 and +1.87% at $\rho$=100), indicating that risk-bounded calibration remains beneficial even when the class count is small. Notably, RBDR-DDIM is largely on par with the default RBDR on CIFAR10-LT (e.g., 81.57% vs. 81.69% at $\rho$=200), suggesting that diffusion-based synthesis becomes more competitive in the low-category regime where conditional generation is easier to fit.

*Table 5.* Top-1 accuracy (%) of various methods on CIFAR10-LT. **Bold** denotes the best result in each column. Underline denotes the best result among feature-space distribution reconstruction methods.

| Method | CIFAR10-LT | | | |
| --- | --- | --- | --- | --- |
| | 200 | 100 | 50 | 10 |
| CE | 65.60 | 70.40 | 74.80 | 86.40 |
| LDAM-DRW | 74.74 | 77.03 | 81.03 | 88.16 |
| BBN | 73.47 | 79.82 | 81.18 | 88.32 |
| GCL | 79.00 | 82.70 | 85.50 | - |
| DisA | 79.10 | 82.80 | 84.90 | 90.50 |
| CE+OTmix | - | 78.30 | 83.40 | 90.20 |
| LA+Focal-SAM | 79.60 | 82.90 | 85.50 | 90.50 |
| IP-DPP | 73.50 | 76.40 | - | - |
| FeatRecon | - | **86.42** | 88.49 | **92.03** |
| *Feature-space distribution reconstruction methods* | | | | |
| LADC | 81.35 | 84.65 | 87.09 | 90.81 |
| FDC | 79.70 | 83.40 | 86.50 | 90.60 |
| FUR | 79.80 | 83.70 | 86.20 | 90.90 |
| RBDR-DDIM (ours) | 81.57 | 85.43 | 88.47 | 91.88 |
| RBDR (ours) | **81.69** | 85.57 | **88.52** | 91.97 |

## E.4. The Effects of $\alpha$ and $\beta$

**Sensitivity to step sizes $\alpha$ and $\beta$.** We study how the mean step size $\alpha$ and the covariance step size $\beta$ affect RBDR on CIFAR100-LT ($\rho$=100), using OT-only calibration as a reference throughout. Figure 5 (left) fixes $\beta$=0.5 and varies $\alpha$. Increasing $\alpha$ from 0 to 0.1 brings a clear gain for both methods, indicating that a small mean correction already captures useful head-to-tail transfer. As $\alpha$ further increases, OT degrades progressively due to overly aggressive mean shifting, whereas RBDR remains substantially more stable, keeping the drop within about 2% over the explored range. This supports the role of rival-aware constraints in limiting harmful drift while retaining the benefit of mean updates.

Figure 5 (middle) fixes $\alpha$=0.3 and varies $\beta$. Both OT and RBDR benefit markedly when $\beta$ increases from 0 to 0.1, suggesting that a mild covariance adaptation is important once tail covariances are poorly estimated. Under this moderate $\alpha$, both methods are relatively insensitive to $\beta$ afterwards, consistent with the fact that covariance shrinkage is most influential

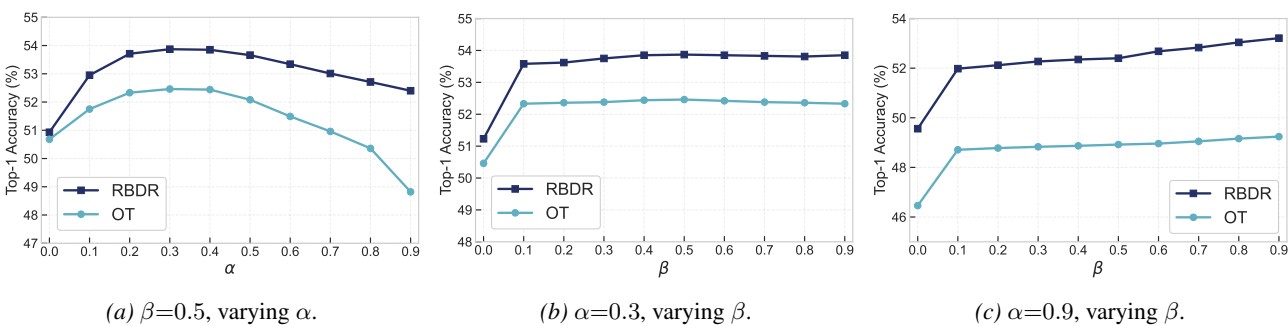

*(a) $\beta$=0.5, varying $\alpha$.*    *(b) $\alpha$=0.3, varying $\beta$.*    *(c) $\alpha$=0.9, varying $\beta$.*

*Figure 5.* Sensitivity of Top-1 accuracy (%) to the mean step size $\alpha$ and covariance step size $\beta$ on CIFAR100-LT ($\rho$=100), comparing OT-only calibration with RBDR.

when class separation is tight.

To further stress the shrinkage effect, Figure 5 (right) sets $\alpha$=0.9 and sweeps $\beta$ again. With stronger mean movement, increasing $\beta$ yields a gradual improvement for both methods, and RBDR enjoys a larger gain than OT, indicating that variance control in rival-aligned subspaces becomes more beneficial when the update is more aggressive. The low-rank shrinkage coefficient $\gamma_{\max}$ plays a similar role in controlling the contraction strength, and its trend is analogous to $\beta$, thus omitted for brevity. Finally, Proposition 3.1 suggests using *moderate* $\alpha$ and $\beta$ to satisfy the sufficient first-order safety condition and prevent overly aggressive calibration. Thus, although RBDR includes several tunable hyperparameters, they are not free knobs: our analysis constrains their feasible ranges and provides practical guidance for stable and consistent updates.

**Sensitivity on ImageNet-LT.** We further evaluate the sensitivity to $\alpha$ and $\beta$ on ImageNet-LT. As shown in Table 6, the results are consistent with the main observations on CIFAR100-LT, but suggest a more conservative mean step for large-scale benchmarks. When $\beta = 0.5$, the best accuracy is obtained around $\alpha = 0.1$, and larger $\alpha$ values lead to gradual degradation. This is expected because ImageNet-LT contains many more classes, making the feature space more crowded and the method more sensitive to aggressive mean shifts. By contrast, when $\alpha = 0.1$, the performance is relatively stable once $\beta \geq 0.1$, indicating that a mild covariance adaptation is sufficient after the tail covariance has been regularized.

*Table 6.* Sensitivity to $\alpha$ and $\beta$ on ImageNet-LT.

| $\alpha\ (\beta = 0.5)$ | 0 | 0.1 | 0.2 | 0.3 | 0.4 | 0.5 | 0.6 | 0.7 | 0.8 | 0.9 |
|---|---|---|---|---|---|---|---|---|---|---|
| Acc. | 53.23 | 55.91 | 55.88 | 55.81 | 55.67 | 55.41 | 55.12 | 54.81 | 54.49 | 54.17 |

| $\beta\ (\alpha = 0.1)$ | 0 | 0.1 | 0.2 | 0.3 | 0.4 | 0.5 | 0.6 | 0.7 | 0.8 | 0.9 |
|---|---|---|---|---|---|---|---|---|---|---|
| Acc. | 53.72 | 55.77 | 55.84 | 55.89 | 55.91 | 55.91 | 55.90 | 55.91 | 55.88 | 55.89 |

### E.5. The Effect of the Number of Rivals $M$

We study the sensitivity to the number of rivals $M$ used in the supportive projection (soft variant; Appendix C.2). As shown in Fig. 6, increasing $M$ yields substantial gains over the OT-only baseline ($M$=0), with performance improving rapidly up to $M \approx 10$ and then saturating. This is expected: the soft objective penalizes only violated constraints via $[-v_{c,k}^{\top}\delta]_+$, so once dominant rivals are covered, most additional constraints become inactive and provide diminishing returns.

Based on this trade-off, we recommend $M \approx 0.1|\mathcal{C}|$ (about 10% of the classes), which is near-optimal on CIFAR100-LT ($|\mathcal{C}|$=100 $\Rightarrow$ $M$=10) and serves as a robust default for larger benchmarks such as ImageNet-LT and iNaturalist 2018. Empirically, in these long-tailed datasets, highly frequent categories typically occupy a small fraction of classes (often around 10%), so a proportional $M$ covers the main confusable rivals while keeping the overhead modest. This is important since the supportive projection cost scales linearly with $M$ and becomes more pronounced at scale.

**Larger-scale validation on ImageNet-LT.** We further verify this trade-off on ImageNet-LT, where the label space is much larger. As shown in Table 7, increasing $M$ from 50 to 100 captures most of the improvement, while using more rivals brings only marginal gains. This trend is consistent with the CIFAR100-LT observation that once the dominant confusable

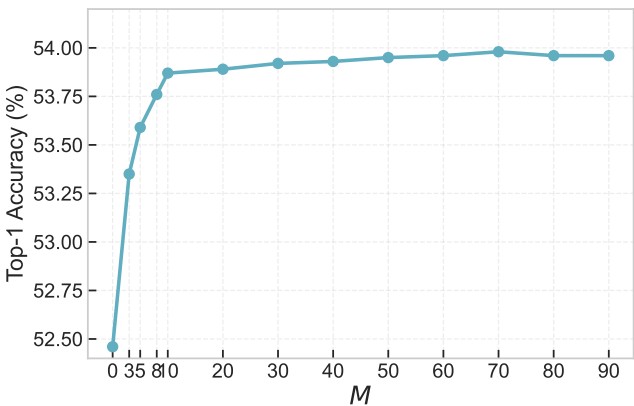

*Figure 6.* Top-1 accuracy of RBDR on CIFAR100-LT as a function of the number of rivals $M$. $M{=}0$ corresponds to OT initialization without rival control.

classes are covered, additional rival constraints are often inactive or redundant. Considering that the supportive projection cost scales linearly with $M$, these results support $M \approx 0.1|\mathcal{C}|$ as a cost-effective choice for large-scale benchmarks.

*Table 7.* Effect of the rival number $M$ on ImageNet-LT.

| $M$ | 50 | 100 | 150 | 200 | 300 | 500 |
|---|---|---|---|---|---|---|
| Acc. | 55.68 | 55.91 | 55.92 | 55.94 | 55.95 | 55.95 |

### E.6. Robustness to $\delta$ Initialization

RBDR starts from an initial head-to-tail update direction $\delta$, and the same weighting scheme also affects the head-aggregated covariance used for shrinkage. Therefore, this study can be viewed as a *plug-in test*: we ask whether RBDR remains effective when the initialization signal is replaced by different head-to-tail transfer rules.

**Setups.** We consider three representative choices for selecting head classes: (i) **Top-$k$** heads with uniform weights (similar to (Yang et al., 2021)); (ii) **Weighted Top-$k$**, where weights are proportional to the cosine similarity between class means (similar to (Wang et al., 2022; Yi et al., 2025)); and (iii) **Top-all**, which uses all head classes. Unless otherwise specified, we set $k{=}10$. In all cases, we apply the same RBDR modules, i.e., supportive projection (SP) followed by low-rank shrinkage (LS).

**Results.** Tables 8 to 10 show that RBDR improves performance under all three initializations on CIFAR100-LT ($\rho{=}100$), ImageNet-LT, and iNaturalist 2018. Notably, starting from Top-all yields larger final gains than Top-$k$ after applying RBDR. A plausible explanation is that Top-all provides a richer pool of head directions: after suppressing rival-aligned components, more informative head-to-tail cues can still be retained safely.

This trend also highlights OT as a principled initializer: its transport weights allocate head information to each tail class in a structured manner, so supportive projection mainly needs to remove a small set of unsafe directions to produce a well-behaved update $\delta$.

*Table 8.* **Top-$k$** initialization ($\delta$ from Top-$k$ heads).

| Method | CIFAR100-LT | ImageNet-LT | iNaturalist 2018 |
|---|---|---|---|
| Top-$k$ | 50.73 | 51.67 | 69.24 |
| + SP | 51.51 | 52.88 | 70.76 |
| + LS | 51.93 | 53.65 | 71.44 |

*Table 9.* **Weighted Top-$k$** initialization (weights $\propto$ mean cosine similarity).

| Method | CIFAR100-LT | ImageNet-LT | iNaturalist 2018 |
|---|---|---|---|
| Weighted Top-$k$ | 50.97 | 51.96 | 69.51 |
| + SP | 51.72 | 53.28 | 70.95 |
| + LS | 52.09 | 53.96 | 71.69 |

*Table 10.* **Top-all** initialization ($\delta$ from all head classes).

| Method | CIFAR100-LT | ImageNet-LT | iNaturalist 2018 |
|---|---|---|---|
| Top-all | 50.75 | 51.71 | 69.27 |
| + SP | 51.69 | 53.08 | 70.96 |
| + LS | 52.15 | 53.94 | 71.73 |

## E.7. Computational Cost

We report the computational cost of the three main components in RBDR on a single NVIDIA RTX 4090 GPU. All three modules are implemented with CUDA acceleration. Since RBDR is an *offline* pipeline applied on frozen backbone features, we exclude backbone training. We also omit classifier fine-tuning time, which is negligible compared with backbone training. Overall, the dominant cost comes from: (i) OT computation, (ii) supportive projection (SP), and (iii) per-class low-rank shrinkage (LS). Results are summarized in Table 11.

**Runtime breakdown.** OT computes the head-to-tail transport plan $T$ once using Sinkhorn ($\varepsilon=2$; $\leq$200 iterations with early stopping), taking 0.135 s on CIFAR100-LT ($\rho=100$), 1.551 s on ImageNet-LT, and 62.28 s on iNaturalist 2018 due to the much larger class set. SP applies rival-aware supportive projection via fully vectorized batched operations; it is negligible on CIFAR100-LT (0.257 ms) and takes 2.729 s on iNaturalist 2018, where we compute SP in mini-batches over tail classes to avoid OOM. LS performs low-rank shrinkage class-by-class; the reported time is measured after completing the loop over all tail classes, yielding 0.115 s (CIFAR100-LT), 1.158 s (ImageNet-LT), and 9.827 s (iNaturalist 2018).

*Table 11.* Runtime (s) of the main modules in RBDR.

| Module | CIFAR100-LT | ImageNet-LT | iNaturalist 2018 |
|---|---|---|---|
| OT | 0.135 | 1.551 | 62.28 |
| SP | 0.000257 | 0.056 | 2.729 |
| LS | 0.115 | 1.158 | 9.827 |

**Full offline cost on iNaturalist 2018.** Beyond the module-level runtime in Table 11, we report the end-to-end offline cost of the default RBDR pipeline on iNaturalist 2018. This measurement includes OT, SP, and LS computation, I/O and orchestration overhead, and looping over tail classes. As shown in Table 12, the full calibration takes 2.2 minutes on a single RTX 4090 workstation, with 10.3 GB peak RAM and 3.5 GB peak GPU memory.

**Cost of RBDR-DDIM.** Table 13 reports the cost of RBDR-DDIM on iNaturalist 2018, measured on a single RTX 4090 GPU. Training is performed on frozen features with batch size 256, taking 275 s per epoch. At inference, we synthesize up to 1,000 samples per tail class, requiring 1,134 s in total. Peak GPU memory is 3.6 GB during training and 3.5 GB during inference. CPU RAM usage is 11.2 GB during training to hold the feature tensor, and 12.3 GB during inference with our streaming implementation, which writes synthesized samples to disk as shards and loads only mini-batches during classifier fine-tuning. Overall, RBDR-DDIM has bounded memory and a moderate one-time offline runtime.

*Table 12.* Full offline cost of the default RBDR pipeline on iNaturalist 2018.

| Dataset | Total RBDR Runtime (min) | Peak RAM (GB) | Peak GPU Mem. (GB) |
|---|---|---|---|
| iNaturalist 2018 | 2.2 | 10.3 | 3.5 |

*Table 13.* Computational cost of RBDR-DDIM.

| Train (s) | Infer (s) | GPU Mem. Train (GB) | GPU Mem. Infer (GB) | RAM Train (GB) | RAM Infer (GB) |
|---|---|---|---|---|---|
| 275 | 1134 | 3.6 | 3.5 | 11.2 | 12.3 |

