# OpenReview forum: "Risk-Bounded Distribution Reconstruction: Stable Statistic Calibration for Long-Tailed Recognition"
_ICML.cc/2026/Conference — ICML 2026 regular_

### Official Review · Reviewer_sZtQ · 2026-02-25

**Soundness:** 3
**Presentation:** 3
**Significance:** 3
**Originality:** 3
**Overall Recommendation:** 4
**Confidence:** 3

**Summary:**

This paper proposes a "Risk-Bounded Distribution Reconstruction" offline statistic calibration framework for long-tailed recognition. Deviating from previous heuristic feature reconstruction methods, this paper approaches the problem by analyzing the confusion risk from "rival" classes and derives safe statistic calibration constraints based on a distribution-free Cantelli surrogate bound and Fisher separation. The proposed method consists of two main modules: (1) mean calibration via Optimal Transport initialization followed by a supportive projection, and (2) covariance calibration through low-rank shrinkage in the rival-induced subspace. This framework is validated on several long-tailed recognition benchmarks as a plug-and-play offline calibration step.

**Compliance With Llm Reviewing Policy:**

Affirmed.

**Final Justification:**

This article is theoretically sound, and I think this article deserves to be recommended for acceptance.

**Key Questions For Authors:**

See Weakness.

**Limitations:**

Yes, the authors have discussed potential risks in the Impact Statement section.

**Strengths And Weaknesses:**

### Strengths
1. The paper steps out of the traditional mindset of merely "increasing tail data diversity" and astutely captures the phenomenon that errors in long-tailed recognition are often concentrated on a few similar "rival" classes. This perspective is quite refreshing and insightful.
2. The theoretical foundation of this paper is highly solid. By establishing a monotonic link between the Fisher ratio and the Cantelli bound, it provides meaningful theoretical support for head-to-tail feature transfer in long-tailed learning.
### Weaknesses
1. The actual performance of the proposed method is somewhat modest. Compared to existing baselines, the performance gains are quite limited.
2. The proposed method is quite complex. Although the authors state that they use a lightweight generative model for efficiency, its introduction still brings extra computational overhead compared to the baselines. The authors should discuss this more thoroughly. Specifically, are these additional computational costs acceptable given the marginal performance improvements?
3. This is a highly theoretical paper that requires readers to have a deep mathematical background. To lower the reading barrier, the authors are strongly encouraged to include a framework overview diagram/figure to help readers better grasp the pipeline and core concepts.
4. Although the authors claim that their modules are "plug-and-play", there are not enough extensive plug-and-play experiments to fully substantiate this claim. Furthermore, for a supposedly "plug-and-play" method, the overall pipeline is overly complex and involves numerous hyperparameters that require manual tuning. This significantly increases the difficulty and threshold for applying this method in practice.
5. In Section E.2, to avoid computing the inverse of the covariance matrix $S_w^{-1}$, the authors adopt an approximation based on the classical Gaussian discriminant/LDA model. The prerequisite for this is the assumption of a "shared within-class covariance" across all classes. However, this assumption is fundamentally invalid in long-tailed data scenarios, where the covariance disparity between classes is massive. In fact, addressing inaccurate and massive tail covariances is one of the exact motivations presented by the authors in the main text. Thus, the logic here appears self-contradictory.


As my expertise does not primarily lie in pure theoretical derivations, my auditing of the theoretical proofs might be limited, hence I set my confidence score to 3.And, If the authors can satisfactorily address my concerns above, I will be happy to raise my score.

---

> ### Author Rebuttal · Authors · 2026-03-30
>
> We sincerely thank you for the careful and constructive review. Below we respond point by point.
>
> **W1. The actual performance gains appear somewhat modest.**
>
> **A1:** Thank you for raising this important point. Our goal is not to claim universal SOTA over all long-tailed methods. The contribution of RBDR is narrower and more principled: to turn heuristic feature-space reconstruction into a rival-aware, safety-constrained offline calibration procedure. Under this scope, the empirical results are still meaningful: the default RBDR consistently improves prior feature-space reconstruction baselines across the three benchmarks, and the gains match the intended behavior of improving tail separability while limiting unsafe drift along rival-aligned directions. This is also supported by the rival-oriented diagnostics and module ablations. We will clarify this positioning in the revision.
>
> **W2. The method is complex, and the DDIM variant adds overhead relative to the marginal gains.**
>
> **A2:** Thank you for raising this important point. The practical cost-benefit trade-off should be judged primarily on the default Gaussian-based RBDR, which is our main practical method and recommendation. Importantly, the default RBDR is a single-round offline post-hoc calibration on frozen features, rather than an iterative end-to-end retraining scheme, and its overhead remains manageable in practice (please also see **Reviewer iB1N (Q1)**). By contrast, RBDR-DDIM is not a required part of the default pipeline, but an exploratory extension showing that the rival-aware calibration framework is not tied to a single Gaussian generator.
>
> **W3. The paper is highly theoretical and should include an overview figure.**
>
> **A3:** We appreciate this suggestion. In the revision, we will add a concise pipeline overview figure to summarize the default workflow (feature extraction → OT-initialized supportive mean calibration with covariance calibration → tail feature synthesis and rival-subspace shrinkage → classifier fine-tuning) and lower the reading barrier.
>
> **W4. The “plug-and-play” claim is not sufficiently substantiated, and the overall pipeline still seems complicated.**
>
> **A4:** Thank you for this helpful comment. Our intended meaning of “plug-and-play” is narrower than “can be inserted unchanged into arbitrary long-tailed methods.” What we mean is that RBDR acts as a structured offline calibration module for the standard two-stage frozen-feature workflow. In this sense, it is a post-hoc add-on to a standard frozen-feature pipeline, rather than a new end-to-end training recipe (please also see **Reviewer 9itz (Q1)**).
>
> We also agree that the practical tuning scope should be stated more clearly. In practice, the main tunable parameters are $\alpha$, $\beta$, and $M$; the internal parameters are kept at fixed defaults (see **Reviewer iB1N (Q2) and Reviewer 9itz (W1/W2)**). We will therefore revise the wording so that the paper presents RBDR more precisely as a structured offline calibration module for standard frozen-feature pipelines.
>
> **W5. The shared-covariance / LDA approximation in Appendix E.2 seems self-contradictory.**
>
> **A5:** We appreciate this important concern. First, the core theory in Sections 2–3 does not assume a globally shared within-class covariance. The main analysis is pairwise and class-specific, based on quantities such as $\Sigma_c$, $\Sigma_k$, and $S_{c,k}=\Sigma_c+\Sigma_k$, precisely to account for heterogeneous and unreliable tail-class covariances. The shared-covariance LDA discussion in Appendix E.2 was intended only as a motivating special case, not as a defining assumption of the main method.
>
> More importantly, there is a distinction between the population-level analytical quantity used in the theory and its direct finite-sample plug-in estimate in practice. In the analysis, the inverse-based direction $S_{c,k}^{-1}(\mu_c-\mu_k)$ is a conceptual rival-sensitive discriminative axis. What becomes problematic in long-tailed practice is its direct empirical estimate $(\hat\Sigma_c+\hat\Sigma_k)^{-1}(\hat\mu_c-\hat\mu_k)$, which can be highly unstable for scarce tail classes because $\hat\Sigma_c+\hat\Sigma_k$ is often ill-conditioned. For this reason, our practical method uses the classifier-weight proxy $w_c-w_k$ as a numerically robust surrogate. The goal is not to claim exact equivalence in the long-tailed regime, but to obtain a stable direction that captures the rival-aware discriminative tendency needed to construct the dangerous subspace. We additionally verified this on the last 10 tail classes of CIFAR100-LT and their nearest rivals: the direct inverse-based directions frequently had extremely large norms (up to the order of $10^6$), and replacing the weight-difference proxy with the direct inverse-based estimate for rival-direction construction caused the final performance to **drop by about 0.5%**. In the revision, we will modify Appendix E.2 to make this distinction explicit.

---

> > ### Author Rebuttal · Reviewer_sZtQ · 2026-04-01
> >
> > I have no more questions and I will raise my rating.

---

> > > ### Author Response · Authors · 2026-04-02
> > >
> > > We sincerely thank you for your careful and detailed reading of our paper and responses. We are grateful that your concerns have been fully resolved, and we greatly appreciate your willingness to update your score.
> > >
> > > Thank you again.

---

### Official Review · Reviewer_9itz · 2026-03-10

**Soundness:** 3
**Presentation:** 3
**Significance:** 3
**Originality:** 3
**Overall Recommendation:** 4
**Confidence:** 3

**Summary:**

This paper proposes Risk-Bounded Distribution Reconstruction (RBDR), an offline statistic calibration framework for long-tailed recognition. The authors observe that existing feature-space reconstruction methods for tail classes often rely on heuristic updates of class statistics, which may inadvertently increase confusion with visually or semantically similar rival classes. To address this issue, the paper introduces a rival-aware theoretical analysis that derives a distribution-free surrogate risk bound based on the Cantelli inequality and establishes its connection to Fisher separation. Guided by this analysis, the proposed method performs risk-bounded mean calibration via supportive projection and covariance calibration with rival-direction variance control. These modules aim to ensure that statistic updates do not increase discriminative risk. Extensive experiment on various long-tailed datasets validate the effectiveness of the proposed method.

**Compliance With Llm Reviewing Policy:**

Affirmed.

**Final Justification:**

The paper presents a well-motivated framework with a solid theoretical basis and fairly comprehensive experimental evaluation, demonstrating consistent improvements on standard long-tailed benchmarks. The analysis from the perspective of rival classes is interesting and provides useful insights. However, the method is relatively complex with multiple components and hyperparameters, and the discussion of related work could be more complete.

During the rebuttal phase, the authors have adequately addressed my concerns and clarified the raised questions. Therefore, I decide to maintain my score at 4.

**Key Questions For Authors:**

- The experiments in this paper are mainly conducted on ResNet-based backbones. However, recent long-tailed recognition works have begun to explore fine-tuning CLIP, such as methods like LIFT [1]. Could the authors discuss whether the proposed method can be applied to such models and whether similar benefits are expected in that setting?

-----

[1] Long-Tail Learning with Foundation Model: Heavy Fine-Tuning Hurts, ICML 2024

**Limitations:**

yes.

**Strengths And Weaknesses:**

**Strengths:**

- The paper introduces a relatively novel perspective by analyzing tail distribution reconstruction from the viewpoint of rival classes, i.e., the classes that a tail category is most likely to be confused with. Based on this observation, the authors develop a rival-aware theoretical analysis of statistic calibration and further design corresponding calibration mechanisms. This perspective provides a more targeted way to analyze and control tail-class confusion in long-tailed recognition.

- The method has a relatively solid theoretical basis. In particular, the paper derives a distribution-free surrogate risk bound using the Cantelli inequality and further connects this surrogate to Fisher separation, which gives useful justification for the proposed statistic calibration strategy.
- The experimental evaluation is fairly comprehensive, covering multiple standard long-tailed benchmarks including CIFAR-LT, ImageNet-LT, and iNaturalist 2018, and the method shows consistent improvements over several feature-space reconstruction baselines. The paper also includes several analysis experiments, such as sensitivity analysis and computational cost, which provide further insight into the effectiveness of the proposed components.



**Weaknesses:**

- The proposed method appears relatively complex, as it consists of several components, including optimal transport initialization, supportive projection for mean calibration, covariance calibration, and rival-subspace variance control. The combination of multiple modules may make the overall framework harder to implement and integrate in practice.
- The proposed framework introduces several hyperparameters related to mean calibration, covariance shrinkage, rival set construction, and variance control. The relatively large number of hyperparameters may make the method more difficult to tune in practice and could limit its ease of adoption.

- More representative long-tailed learning methods, such as PaCo [1], GPaCo [2], SADE [3], DirMixE [4], should be included in the related work for a more comprehensive review.
- There are some typos. For example, in line 408, the term "optical benchmarks" seems inappropriate and should likely be replaced with "visual benchmarks".

-----

[1] Parametric contrastive learning, ICCV 2021

[2] Generalized Parametric Contrastive Learning, TPAMI 2023

[3] Self-Supervised Aggregation of Diverse Experts for Test-Agnostic Long-Tailed Recognition, NeurIPS 2022

[4] Harnessing Hierarchical Label Distribution Variations in Test Agnostic Long-tail Recognition, ICML 2024

---

> ### Author Rebuttal · Authors · 2026-03-30
>
> We sincerely thank you for your thoughtful and constructive feedback. Below, we respond point by point below.
>
> **W1. The method appears relatively complex and may be harder to implement / integrate in practice.**
>
> **A1:** We appreciate this concern. Although the framework is modular, the default RBDR is still a single-round offline post-hoc calibration on frozen features, rather than an iterative end-to-end retraining scheme. Although the method is presented in several theory-aligned components, its practical implementation is relatively simple: after feature extraction, the default pipeline is executed sequentially in one offline pass and does not introduce repeated optimization loops over the backbone.
>
> Its overhead also remains moderate in practice. Please see our response to **Reviewer iB1N (Q1)** for the new result on iNaturalist 2018: the full default RBDR calibration finishes in 2.2 min with 10.3 GB RAM and 3.5 GB GPU memory on a single RTX 4090 workstation, in addition to the module-level breakdown already reported in Table 9 / Appendix E.7. These results suggest that, although RBDR contains multiple modules, its practical execution cost remains manageable.
>
> **W2. The framework introduces several hyperparameters and may be difficult to tune in practice.**
>
> **A2:** We agree this should be clarified better. In practice, the main tunable parameters are **$\alpha$, $\beta$, and $M$**, and they are directly tied to the RBDR theory: $\alpha$ controls the magnitude of mean calibration, $\beta$ controls the strength of covariance / variance calibration, and $M$ determines how many dominant rivals are included in the rival-aware constraints. The remaining internal parameters, including those in **supportive projection** and **low-rank shrinkage**, are kept at fixed default settings in our implementation (see Appendix E.2). In our experience, these parameters show stable behavior and have much smaller practical impact than the main theoretical parameters $α$, $β$, and $M$, so we keep them unchanged across datasets rather than tuning them separately.
>
> Please also see our response to **Reviewer iB1N (Q2)** for the new ImageNet-LT sensitivity study. The main pattern is clear and consistent with Appendix E.4/E.5: $\alpha$ works best around 0.1, performance is not very sensitive to $\beta$ once $\beta \ge 0.1$, and performance is nearly saturated once $M$ reaches a moderate range, consistent with **$M \approx 0.1|C|$**. In practice, we recommend starting from small $(\alpha,\beta)$ and $M \approx 0.1|C|$, and only adjusting if validation performance indicates clear under- or over-correction.
>
> **W3/W4. Related work and wording issues.**
>
> **A3:** Thank you for pointing these out. We will expand the related work to include representative methods such as PaCo, GPaCo, SADE, and DirMixE, and we will carefully proofread the manuscript and fix wording issues (e.g., “optical benchmarks” → “visual benchmarks”).
>
> **Q1. Can the proposed method be applied to CLIP / LIFT-style settings?**
>
> **A4:** Thank you for this helpful question. RBDR is backbone-agnostic in principle, since it operates on frozen feature embeddings and performs post-hoc class-statistics calibration before classifier refinement. To examine this more directly, we conducted preliminary experiments on CIFAR100-LT using frozen **ViT-B/16** features from **CLIP[1]** and **DINOv2[2]** with an MLP classifier, and observed consistent gains after applying RBDR: **CLIP+MLP 47.5/49.6 → 64.4/66.9** and **DINOv2+MLP 70.7/76.2 → 76.5/82.1** for imbalance ratios 200/100. This suggests that the rival-aware calibration idea is not limited to ResNet backbones.
>
> For more established long-tailed learning pipelines such as LIFT, however, we would be more cautious. The current RBDR formulation is built for an offline post-hoc calibration stage with a frozen backbone, so it is not directly formulated as an online training module. Still, we believe the same rival-aware principle can be extended to online training in a lighter form, e.g., as a regularization term. As for whether similar benefits should be expected, our view is that the rival-aware intuition should still be helpful, but likely in a different way. For example, LIFT combines several effective ingredients, including LA-based training, semantic-aware initialization, and test-time ensembling. In such a setting, an online RBDR-style component would more likely act as a complementary regularizer rather than the dominant source of gain. Accordingly, we would expect the same rival-aware principle to remain beneficial, although the magnitude and form of the gain would likely differ from the current offline RBDR setting.
>
> [1] Learning transferable visual models from natural language supervision. ICML 2021
>
> [2] Dinov2: Learning robust visual features without supervision. TMLR 2024

---

> > ### Author Rebuttal · Reviewer_9itz · 2026-04-03
> >
> > Thank you for the detailed and helpful response. It has addressed most of my concerns and clarified the key questions I raised. I will maintain my original positive score.

---

> > > ### Author Response · Authors · 2026-04-03
> > >
> > > Thank you very much for your careful reading and positive acknowledgement. We sincerely appreciate your encouraging feedback and continued support.

---

### Official Review · Reviewer_iB1N · 2026-03-12

**Soundness:** 3
**Presentation:** 3
**Significance:** 3
**Originality:** 3
**Overall Recommendation:** 4
**Confidence:** 3

**Summary:**

This paper proposes Risk Bounded Distribution Reconstruction for long tailed recognition. Long tailed recognition has a big problem with extreme class imbalance. Existing methods use heuristic rules to reconstruct features and often hurt the separation between different classes. To solve this, the authors propose a new offline framework to calibrate statistics safely. The method updates the class mean by projecting candidate directions onto a supportive set. It also controls the covariance by reducing the dispersion in a dangerous rival subspace. The experiments show that this new method improves the accuracy and stability on multiple long tailed datasets.

**Compliance With Llm Reviewing Policy:**

Affirmed.

**Final Justification:**

The author has met my requirements, so I choose to maintain my score.

**Key Questions For Authors:**

- Table 9 shows that the Optimal Transport and Low Rank Shrinkage steps take extra time. Can you provide the exact memory and time costs for training on much larger datasets? If the costs are very high, the method will be hard to use in real applications.
- The method introduces several new parameters like the mean step size and the covariance step size. How sensitive is the final accuracy to these parameters on a completely new dataset?
- Table 1 shows that the complex diffusion variant often performs worse than the simple default method. Why do we need the complex diffusion model if the simple Gaussian method works better?

**Limitations:**

See weaknesses.

**Strengths And Weaknesses:**

Strengths.
- The paper introduces a new idea to use rival directions to guide statistic calibration. This provides a safe and clear way to reconstruct distributions for long tailed recognition.
- The experiments on large datasets like ImageNet LT and iNaturalist show consistent improvements over baseline methods.

Weaknesses.
- The method uses optimal transport and multiple rival constraints. This mathematical process needs more computation time and memory than simple heuristic methods during the offline stage.
- The framework introduces several new parameters. For example, users need to choose the number of rivals and the step sizes. This makes the method hard to tune for completely new datasets.
- The paper proposes a complex diffusion variant called RBDR DDIM. However, the experiments show that this complex variant does not always perform better than the simple Gaussian method. This makes the diffusion part less practical.

---

> ### Author Rebuttal · Authors · 2026-03-30
>
> **We sincerely thank you for your thoughtful and constructive feedback. Below, we respond to your comments point by point.**
>
> **Q1. Computation time and memory cost.**
>
> **A1:** We agree that the practical cost should be made explicit. In addition to the per-module runtime already reported in Table 9, we further measured the total runtime, peak RAM, and peak GPU memory of the full default RBDR pipeline on iNaturalist 2018, using a single RTX 4090 workstation. This reported cost reflects the full RBDR procedure in practice, including OT/SP/LS computation, I/O, orchestration overhead, and looping over tail classes.
>
> | Dataset | Total RBDR Runtime (min) | Peak RAM (GB) |  Peak GPU Mem. (GB)  |
> |:-------:|:------------------------:|:-------------:|:--------------------:|
> | iNaturalist 2018 | 2.2 | 10.3 |         3.5          |
>
> Together with the module-level breakdown in Table 9, this result suggests that the additional offline cost of RBDR over simpler heuristic methods remains manageable even on the largest benchmark in this paper. Since RBDR is an offline post-hoc calibration stage rather than an end-to-end retraining procedure, we believe that the full calibration cost reported here provides a more direct practical picture. We will add this result to Appendix E.7 and clarify this point more explicitly in the revision.
>
> **Q2. Hyperparameter sensitivity on a new dataset.**
>
> **A2:** Thank you for raising this point. To address this more directly, we additionally performed a simple sensitivity study on **ImageNet-LT** for $\alpha$, $\beta$, and the rival number $M$.
>
> **(i). $β=0.5$, varying $α$.**
>
> |     | 0     | 0.1   | 0.2   | 0.3   | 0.4   | 0.5   | 0.6   | 0.7   | 0.8   | 0.9   |
> |:---:|-------|-------|-------|-------|-------|-------|-------|-------|-------|-------|
> | $α$ | 53.23 | 55.91 | 55.88 | 55.81 | 55.67 | 55.41 | 55.12 | 54.81 | 54.49 | 54.17 |
>
> **(ii). $α=0.1$, varying $β$.**
>
> |     | 0     | 0.1   | 0.2   | 0.3   | 0.4   | 0.5   | 0.6   | 0.7   | 0.8   | 0.9   |
> |:---:|-------|-------|-------|-------|-------|-------|-------|-------|-------|-------|
> | $β$ | 53.72 | 55.77 | 55.84 | 55.89 | 55.91 | 55.91 | 55.90 | 55.91 | 55.88 | 55.89 |
>
> **(iii). Varying the rival number $M$**
>
> |     | 50    | 100   | 150   | 200   | 300   | 500   |
> |:---:|-------|-------|-------|-------|-------|-------|
> | $M$ | 55.68 | 55.91 | 55.92 | 55.94 | 55.95 | 55.95 |
>
> The main observations are:
>
> - **$\alpha$** works best around **0.1** on ImageNet-LT. Compared with CIFAR100-LT, ImageNet-LT is more crowded in feature space due to the much larger number of classes, so performance is naturally more sensitive to large mean shifts.
> - Under small $\alpha$, the method is not very sensitive to $\beta$ once $\beta \ge 0.1$, which is consistent with the trend shown in Fig. 4.
> - For the rival number **$M$**, performance changes only marginally once **$M \ge 100$**, which is broadly consistent with our practical rule of thumb **$M \approx 0.1|C|$**.
>
> These results suggest that, although RBDR introduces explicit parameters, they are not arbitrary knobs. Their roles are interpretable, and in practice their tuning can be guided by simple empirical heuristics together with the theoretical intuition discussed in the paper.
>
> **Q3. Why we keep RBDR-DDIM if the Gaussian default works better.**
>
> **A3:** Thank you for this helpful question. The default Gaussian-based RBDR is our main practical method and recommended choice. In the current benchmarks, RBDR-DDIM does not outperform this default, so we do not present it as a better practical alternative. We keep it as an exploratory extension for two reasons. First, it helps clarify that the rival-aware calibration framework is not tied only to a Gaussian sampling backend, but can also be combined with a more flexible latent generator. Second, it helps explain the empirical boundary of our method: on the current visual long-tailed benchmarks, moment-matched Gaussian sampling is already a strong default, while class-conditional feature diffusion is harder to train reliably under many classes and severe per-class sparsity, which may reduce its practical advantage here. Please also see our response to **Reviewer M4iv (Q1)** for a more detailed clarification of this role and why RBDR-DDIM does not outperform the default variant in the present setup.

---

> > ### Author Rebuttal · Reviewer_iB1N · 2026-04-03
> >
> > Thank you for your rebuttal and for addressing my concerns. I'd like to maintain my score.

---

> > > ### Author Response · Authors · 2026-04-04
> > >
> > > Thank you very much for your careful reading and positive acknowledgement. We sincerely appreciate your feedback and your decision to maintain your positive score.

---

### Official Review · Reviewer_M4iv · 2026-03-13

**Soundness:** 3
**Presentation:** 3
**Significance:** 2
**Originality:** 3
**Overall Recommendation:** 4
**Confidence:** 4

**Summary:**

This paper proposes Risk-Bounded Distribution Reconstruction (RBDR), an offline statistic calibration framework for two-stage long-tailed recognition. The method is motivated by the limitation of prior feature-space reconstruction approaches, whose heuristic tail-statistic updates can harm multi-class separability when tail estimates are unreliable. The paper develops a rival-aware theoretical framework based on a Cantelli surrogate and Fisher separation, and instantiates it through supportive projection for mean calibration and covariance calibration with rival-aware dispersion control in rival-induced subspaces. Experiments on CIFAR-LT, ImageNet-LT, and iNaturalist 2018 show consistent gains over prior feature-space reconstruction methods, and further analyses support the proposed theory and design choices.

**Compliance With Llm Reviewing Policy:**

Affirmed.

**Ethical Review Flag:**

Flag this paper for an ethics review.

**Ethics Expertise Needed:**

["Other Expertise"]

**Key Questions For Authors:**

Could the authors provide further analysis, or additional experiments, to clarify when RBDR-DDIM is expected to be beneficial and why it does not outperform the default RBDR in the current setup?

**Limitations:**

Yes.

**Strengths And Weaknesses:**

Strengths:

- The paper provides a solid and well-motivated theoretical analysis, which offers meaningful support for the proposed method and helps clarify why the design is effective.
- The paper is well written, with a clear presentation and strong overall organization, making the technical ideas easy to follow.
- The proposed method is conceptually clean and practically appealing: it turns heuristic head-to-tail reconstruction cues into a controllable offline calibration procedure.

Weaknesses:

- The empirical value of the diffusion-based variant RBDR-DDIM is not yet fully clear. It does not outperform the default RBDR in the reported experiments, despite incurring additional offline training and sampling cost. Additional experiments or further analysis would help clarify the scenarios in which this extension may be useful and better justify its practical value.
- While the ablation study validates the contributions of the two main modules, it provides limited analysis of the role of other components in the overall framework, such as the Optimal Transport initialization. Additional ablations on these design choices would make the empirical justification of the full pipeline more complete.

---

> ### Author Rebuttal · Authors · 2026-03-25
>
> We sincerely appreciate your encouraging and thoughtful feedback on our work. Below, we respond to your comments point by point.
>
> **Q1. The practical value of RBDR-DDIM, and when it may be beneficial.**
>
> **A1:** We agree that the current draft does not position RBDR-DDIM clearly enough. Our main practical method and recommendation is the default Gaussian-based RBDR pipeline. At the same time, we do not view RBDR-DDIM as a filler component. Its role is methodological rather than purely practical: it shows that the rival-aware calibration framework is **not restricted to a single Gaussian generator**, and can also be paired with a more flexible latent generative backend.
>
> We believe this extension is well motivated because diffusion-based long-tailed feature generation is still a relatively open direction. Existing latent-space diffusion approaches for long-tailed recognition mainly show that directly training a diffusion model on imbalanced latent features can provide some gains [1], but they do not explicitly address how to reduce head-class bias or how to make the generative process compatible with rival-aware calibration. By contrast, our RBDR-DDIM is designed with this issue in mind: it uses **head-guided skeleton injection** and **mean-centered residual modeling** to make the generative module better suited to long-tailed feature reconstruction and to integrate it into the broader RBDR framework.
>
> At the same time, in the current benchmarks, the default RBDR consistently performs better than RBDR-DDIM. A plausible explanation is that the evaluated datasets are **visual benchmarks**, where moment-matched Gaussian sampling is already a strong baseline in frozen feature space [2], especially when combined with our conservative rival-aware calibration. In contrast, diffusion-based synthesis introduces a more flexible generator, but in many-class, severely sparse long-tailed settings, class-conditional feature diffusion is harder to fit reliably. Moreover, under strong class imbalance, diffusion training can become biased toward head classes, which makes tail-class generation less accurate and less stable [3]. As a result, the added flexibility may even increase dispersion along rival-aligned directions, partially offsetting the benefit of conservative calibration.
>
> Therefore, although the default RBDR remains stronger and more practical in the current setting, we believe RBDR-DDIM is still a useful exploratory extension: it demonstrates that rival-aware calibration can be coupled with a richer latent generative backend, which may support future improvements for more complex tail-shape modeling. At the same time, such a more flexible backend may be especially attractive in settings where the tail feature shape departs more substantially from a single-Gaussian approximation, such as domains with more complex feature geometry (e.g., SAR or medical imaging). We do not claim that this is established by the current benchmarks; rather, it is the motivation for including DDIM as an exploratory extension. We will revise the paper to make this positioning clearer.
>
> **Q2. The role of OT initialization.**
>
> **A2:** We appreciate this point. Related evidence is  provided in **Appendix E.6**, where we replace the initial head-to-tail transfer rule with **Top-k** (similar to [4]), **Weighted Top-k** (similar to [5]), and **Top-all**, while keeping the same RBDR modules (supportive projection + low-rank shrinkage). Across CIFAR100-LT, ImageNet-LT, and iNaturalist 2018, RBDR consistently improves performance under all three alternatives. In particular, relative to the corresponding initialization baseline, adding **SP+LS** yields gains of about **+1.2\% to +1.4\%** on CIFAR100-LT, **+2.0\% to +2.3\%** on ImageNet-LT, and **+2.2\% to +2.5\%** on iNaturalist 2018.
>
> This indicates that OT is not the sole source of the gain; rather, it serves as a principled structured initializer that often provides a cleaner candidate direction, while the main improvements come from the subsequent rival-aware calibration modules. We will surface this evidence more clearly in the revision.
>
> [1] Han P, et al. Latent-based diffusion model for long-tailed recognition. CVPR Workshop 2024
>
> [2] Du C, et al. Probabilistic Contrastive Learning for Long-Tailed Visual Recognition. TPAMI 2024
>
> [3] Zhang T, et al. Long-tailed diffusion models with oriented calibration. ICLR 2024
>
> [4] Yang S, et al. Bridging the gap between few-shot and many-shot learning via distribution calibration. TPAMI 2021
>
> [5] Yi L, et al. Geometry of Long-Tailed Representation Learning: Rebalancing Features for Skewed Distributions. ICLR 2025

---

> > ### Author Rebuttal · Reviewer_M4iv · 2026-04-04
> >
> > I thank the authors for their clear clarifications. All my concerns have been well addressed, and I will maintain my initial score.

---

> > > ### Author Response · Authors · 2026-04-04
> > >
> > > Thank you very much for your careful reading and positive acknowledgement. We sincerely appreciate your encouraging feedback and your continued support of our paper.

---

### Decision · Program_Chairs · 2026-04-30

**Decision:**

Accept (regular)

**Comment:**

## 1. Summary
This paper proposes **Risk-Bounded Distribution Reconstruction (RBDR)**, an offline statistic calibration framework for long-tailed recognition. The key idea is to make feature-space reconstruction safer and more controllable by calibrating tail-class means and covariances with respect to rival classes, rather than relying on purely heuristic updates. The method combines supportive mean projection and rival-subspace covariance control within a two-stage long-tailed pipeline, and is presented as a plug-and-play offline calibration module.

## 2. Reviewer evaluation and concerns
The reviewers were uniformly positive overall. They found the paper well motivated, technically solid, and clearly written. A major strength noted across reviews is the **rival-aware perspective**, which provides a principled way to analyze why heuristic tail reconstruction can hurt class separability. Reviewers also appreciated the theoretical grounding through the Cantelli-based surrogate risk analysis and Fisher-separation connection, as well as the consistent empirical improvements on CIFAR-LT, ImageNet-LT, and iNaturalist.

The main concerns were relatively limited in scope:
- the **practical value of the DDIM-based variant** was unclear, since it did not outperform the default Gaussian-based version while adding cost;
- the method contains **multiple modules and hyperparameters**, so reviewers requested more discussion of implementation cost, sensitivity, and practical usability;
- some reviewers wanted clearer positioning of the **plug-and-play claim**, additional clarification on the role of OT initialization, and minor improvements to related work coverage and presentation.

## 3. Discussion
The rebuttal addressed these concerns well. The authors clarified that the **default Gaussian-based RBDR** is the main practical method, while **RBDR-DDIM** should be viewed as an exploratory extension showing that the framework is not tied to a Gaussian backend. They also added concrete runtime and memory measurements, showing that the offline cost is manageable in practice, and provided additional hyperparameter sensitivity analysis indicating that the method is reasonably stable with interpretable tuning knobs. The rebuttal further clarified that the main gains do not come solely from OT initialization, but from the subsequent rival-aware calibration modules.

Overall, the paper is technically sound, well executed, and meaningfully differentiated from prior heuristic reconstruction methods. While the DDIM extension is not yet practically compelling, this does not detract from the value of the main contribution. The paper offers a principled and empirically supported calibration framework that should be of interest to the long-tailed recognition community.